# RETHINKING THE SHAPE CONVENTION OF AN MLP

## ABSTRACT

Multi-layer perceptrons (MLPs) conventionally follow a narrow-wide-narrow design where skip connections operate at the input/output dimensions while processing occurs in expanded hidden spaces. We challenge this convention by proposing wide-narrow-wide (Hourglass) MLP blocks where skip connections operate at expanded dimensions while residual computation flows through narrow bottlenecks. This inversion leverages higher-dimensional spaces for incremental refinement while maintaining computational efficiency through parameter-matched designs. Implementing Hourglass MLPs requires an initial projection to lift input signals to expanded dimensions. We propose that this projection can remain fixed at random initialization throughout training, enabling efficient training and inference implementations. We evaluate both architectures on generative tasks over popular image datasets, characterizing performance-parameter Pareto frontiers through systematic architectural search. Results show that Hourglass architectures consistently achieve superior Pareto frontiers compared to conventional designs. As parameter budgets increase, optimal Hourglass configurations favor deeper networks with wider skip connections and narrower bottlenecks—a scaling pattern distinct from conventional MLPs. Our findings suggest reconsidering skip connection placement in modern architectures, with potential applications extending to Transformers and other residual networks.

## 1 INTRODUCTION

Multi-layer perceptrons (MLPs) are classical neural network building blocks with a well-established architectural convention. A typical MLP block expands from an input dimension to a wider hidden dimension, then contracts back to an output dimension, resulting in a "narrow-wide-narrow" shape. This expansion allows the network to perform complex transformations in the higher-dimensional hidden space. The feedforward layer in a transformer typically has a hidden dimension 2 to 4 times larger than the token dimension (Vaswani et al., 2017; Jiang et al., 2023).

Beyond improving gradient flow (He et al., 2016b), skip connections enable incremental learning where networks refine representations through additive corrections rather than complete transformations. When applied to MLPs, the conventional approach maintains narrow-wide-narrow blocks where skip connections operate at the narrower input/output dimensions.

However, this convention constrains all residual updates to operate with the input dimensions. In this paper, we challenge this very convention, and hypothesize that *performing incremental improvement is more effective at higher dimensionality*. We thus propose to invert the shape of the MLP when an MLP is accompanied by a skip connection, i.e. taking a "wide-narrow-wide" (Hourglass) shape. This design maintains the skip connection at the expanded latent dimension while residual computations flow through a narrow bottleneck instead. Our hypothesis is motivated by theoretical insights suggesting that higher-dimensional spaces provide richer feature representations for residual learning, potentially enabling more effective incremental refinements than updates constrained to narrow dimensions.

Implementing wide-narrow-wide MLPs requires lifting input signals to expanded dimensions via linear projection. While conventional practice trains this projection end-to-end, we hypothesize that fixed random projections—inspired by reservoir computing—achieve comparable performance when expansion factors are large. The advantages of such a fixed random input projection can offset the additional burden of having to carry one more matrix-vector computing layer.

To test our hypothesis empirically, we conduct architectural comparisons between conventional ("narrow–wide–narrow") and Hourglass ("wide–narrow–wide") MLP stacks. We evaluate both

architectures on generative tasks, including generative classification, denoising, and super-resolution on MNIST, as well as denoising and super-resolution on ImageNet-32 images. Through systematic architectural search, we characterize the performance–parameter count Pareto frontiers for both designs. Our results demonstrate that Hourglass architectures consistently achieve superior Pareto frontiers compared to conventional designs, even when accounting for the additional parameters in the input projection layer. Furthermore, our ablation studies confirm that the linear input projection can indeed remain fixed at its random initialization with negligible impact on performance, validating both our architectural hypothesis and our parameter-efficient design choice.

Breaking from the conventional expand-then-contract MLP paradigm opens previously unexplored architectural trade-offs. Our experiments reveal that as parameter budgets increase, Pareto-optimal Hourglass architectures consistently favor deeper networks with wider skip connections and narrower bottleneck dimensions—a scaling pattern distinct from conventional MLPs.

Our contributions are:

- We propose inverting the conventional narrow-wide-narrow paradigm to a wide-narrow-wide (Hourglass) MLP design, with an input projection to lift natural signal to the wide dimension.

- We propose that that the required input projection can be fixed at random initialization with negligible performance impact, enabling efficient implementations of wide-narrow-wide MLPs.

- Through empirical validation on generative tasks, we show that the wide-narrow-wide design consistently leads to a superior Pareto frontiers compared to the conventional design.

- Our experiments reveal that Pareto-optimal Hourglass architectures consistently favor deeper networks with wider skip connections and narrower bottleneck dimensions as the parameter count increases.

Supported by the results, we believe that our intuition extends beyond MLPs to other skip-connected architectures including Transformers and Vision Transformers—we discuss these broader implications in our Future Work section.

## 2 BACKGROUNDS AND RELATED WORKS

### 2.1 SKIP CONNECTIONS AND INCREMENTAL IMPROVEMENT IN DEEP NETWORKS

Skip connections, introduced in ResNets (He et al., 2016a), originally addressed gradient flow problems in deep networks but also enable a distinct computational paradigm. Rather than learning complete transformations, residual blocks learn correction terms: a block computes $y = x + \Delta F(x)$, where $\Delta F(x)$ represents a learned correction to the input $x$. This formulation allows each layer to contribute incremental improvements to the evolving representation, enabling effective training of very deep architectures.

This incremental refinement principle has become fundamental across diverse modern architectures. Transformers (Vaswani et al., 2017) apply residual connections twice per block—once for self-attention and once for feed-forward processing—with each sublayer contributing additive refinements. Generative models exemplify this principle explicitly: diffusion models learn denoising steps $x_{t-1} = x_t + \epsilon_\theta(x_t, t)$ (Ho et al., 2020), while flow matching models integrate along learned vector fields $\frac{dx}{dt} = v_\theta(x, t)$ (Lipman et al., 2023). The common thread across these architectures is the preference for small, targeted corrections over complete transformations.

### 2.2 MLP BLOCKS AND SKIP CONNECTION PLACEMENT

Multi-layer perceptrons (MLPs) serve as a canonical case study for skip connection placement. A standard MLP block with skip connections follows the pattern:

$$x_{i+1} = x_i + W_2\sigma(W_1\text{norm}(x_i)) \tag{1}$$

where $x_i, x_{i+1} \in \mathbb{R}^{d_x}$, $W_1 \in \mathbb{R}^{d_h \times d_x}$, $W_2 \in \mathbb{R}^{d_x \times d_h}$, and by common convention, $d_h > d_x$.

This creates in a "narrow-wide-narrow" computational graph: the input dimension $d_x$ expands to the hidden dimension $d_h$, then contracts back to $d_x$ to match the skip connection.

MLP has been embedded in various modern neural network architectures. When instantiated, the skip connection connects to a narrower dimension $d_x < d_h$. For instance, the original transformer

used $d_h = 4d_x$ (Vaswani et al., 2017). Modern language models typically employ expansion $d_h/d_x$ between 2-4 (Jiang et al., 2023; Grattafiori et al., 2024) in their feedforward section.

## 2.3 Theoretical Foundations for High-Dimensional Representations

Several theoretical frameworks suggest that operations in higher-dimensional spaces offer computational advantages.

**Information Preservation via Random Projections** Multiple fields demonstrate that random up-projections preserve essential information regardless of the specific projection used. Reservoir computing employs fixed random input projections in Echo State Networks (Jaeger, 2001), while random features show that any appropriately distributed projection can approximate shift-invariant kernels (Rahimi & Recht, 2007). The Johnson-Lindenstrauss lemma formalizes this principle: random matrices satisfying basic distributional properties preserve geometric structure with high probability (Johnson & Lindenstrauss, 1984). Compressive sensing provides additional theoretical support. Under sparsity assumptions, signals can be recovered from remarkably few random measurements, provided sufficient ambient dimensionality (Candès & Tao, 2005; Donoho, 2006).

The shared insight is that, as long as they satisfy appropriate distributional properties, random projections to higher dimensions preserve information structure while being largely invariant to the specific projection matrix chosen —whether Gaussian, Rademacher, or sparse (Achlioptas, 2003).

**Linear Separability in High Dimensions** Cover's theorem (Cover, 1965) demonstrates that projecting data into sufficiently high-dimensional spaces increases the probability of linear separability. Among kernel methods, Support Vector Machines implicitly operate in high-dimensional feature spaces through the kernel trick, while random feature approximations (Rahimi & Recht, 2007) show that wider representations can approximate complex functions with simpler operations.

## 2.4 Related Work

While we focus on MLPs, it is worth noting that several non-MLP architectures already place skip connections at the widest parts of their computational graphs, though without the intentional dimensional expansion that we hypothesize benefits our proposed wide-narrow-wide MLP design.

**U-Net** architectures (Ronneberger et al., 2015) place skip connections between corresponding layers in encoder-decoder networks, effectively connecting at the widest feature map dimensions before spatial downsampling. The skip connections preserve detailed spatial information at full resolution while processing occurs at coarser scales.

**Mixture-of-Experts** (MoE) architectures (Shazeer et al., 2017; Zhang et al., 2022), when routing inference through a small number of active experts, can be viewed as temporarily creating a wide-narrow-wide computational pattern. Similarly, **LoRA** (Hu et al., 2022) — a parameter-efficient fine-tuning (PEFT) method — appends additional wide-narrow-wide paths to any weight matrix.

However, because these architectures operate at naturally occurring wide dimensions rather than artificially expanded feature spaces, they are not directly comparable to the wide-narrow-wide MLP proposed in this work.

## 3 Wide−narrow−wide incremental−improving MLP

We propose inverting the conventional narrow-wide-narrow MLP design to create wide-narrow-wide (hourglass) blocks. Based on the theoretical foundations discussed in Section 2.3, we hypothesize that architectures with skip connections operating at higher dimensions may enable more advantageous incremental refinement. Under the constraint of maintaining comparable parameter count, this architectural change results in individual MLP blocks with the wide-narrow-wide shape, as illustrated in Fig. 1(a). Skip connections preserve information at the wider dimension while the residual path computes incremental improvement through a narrow bottleneck. This design offers additional architectural flexibility: by using narrower bottleneck dimensions, one can construct deeper networks while maintaining the same parameter budget.

### 3.1 Wide−narrow−wide MLP

We describe a network whose core consists of purely wide-narrow-wide MLP. Such a network is built on three distinct stages.

**Input-to-latent projection.** The input signal $x \in \mathbb{R}^{d_x}$, which can be a natural signal, is first projected to the latent space of $d_z$ dimensions via an *input projection*:

$$z_0 = W_{\text{in}}x, \quad W_{\text{in}} \in \mathbb{R}^{d_z \times d_x}. \tag{2}$$

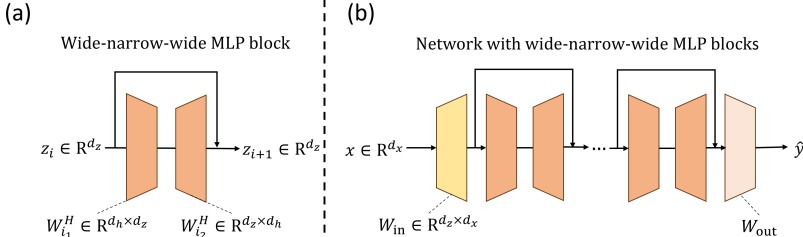

Figure 1: (a) Illustration of a wide-narrow-wide MLP block. The two endpoints $z_i$ and $z_{i+1}$ have a higher dimensionality compared to the hidden $h_i$. Skip connection thus connects two high–dimensional endpoints, rather than two low-dimensional ones in existing convention. Components that do not depend on dimensionality (e.g., normalization, element-wise nonlinearity) are omitted for clarity. (b) Illutration of a full network whose core is a stack of wide-narrow-wide MLP blocks. An input projection network $W_{\text{in}}$ is required to adapt the input dimensionality of $x$ to the dimensionality of the latent $z$. An output projection network $W_{\text{out}}$ is used to adapt to the desired task.

For adapting input signal to a wide-narrow-wide MLP, we consider expansive (up) projection, $d_z > d_x$. When we compare to a network of conventional narrow-wide-narrow MLPs, we follow the common practice of injecting the input signal directly into an MLP, skipping this input projection.

**A stack of MLP blocks.** For block $i = 0, 1, \ldots, L-1$, the incremental improvement is computed and applied in the high–dimensional space:

$$z_{i+1} = z_i + W_{i_2}^H \, \sigma_i\big(W_{i_1}^H \, \text{norm}(z_i)\big), \quad W_{i_1}^H \in \mathbb{R}^{d_h \times d_z}, \quad W_{i_2}^H \in \mathbb{R}^{d_z \times d_h}. \tag{3}$$

If $d_z > d_h$, the MLP is of the wide-narrow-wide type. Conventional MLP has $d_z < d_h$.

**Output conversion.** At the output of the $L$ residual blocks, an additional output network $W_{\text{out}}$ shall be used to convert the last latent $z_L$ into the format demanded by the desired task. For instance, for a training objective aiming to evolve one noised image to a prototypical one, a linear projection $W_{\text{out}} \in \mathbb{R}^{d_x \times d_z}$ can be used:

$$\hat{y} = W_{\text{out}} z_L. \tag{4}$$

If one is interested in only the class tag among $C$ classes of the input $x$, a linear projection $W_{\text{out}} \in \mathbb{R}^{C \times d_z}$ followed by a softmax operation for a distribution over $C$ classes can be applied,

$$\hat{y} = \text{softmax}(W_{\text{out}} z_L). \tag{5}$$

We note that during pretraining, a network with only the input-to-latent projection and the residual blocks can directly learn to predict the optimal output latent. Post-training, an output conversion network can be augmented and then finetuned end-to-end on task-specific data.

### 3.2 Input-to-Latent Projection Strategy

Conventional practice trains the input projection $W_{\text{in}} : \mathbb{R}^{d_x} \to \mathbb{R}^{d_z}$ end-to-end with the rest of the network. However, based on the theoretical foundations discussed in Section 2.3, we propose an alternative approach: using a fixed random projection matrix that remains unchanged throughout training.

We hypothesize that when the expanded dimension $d_z$ is sufficiently larger than the input dimension $d_x$, the performance gap between a randomly initialized projection and a learned one becomes unnoticeable. This hypothesis is motivated by results from reservoir computing, random features, and compressive sensing, which demonstrate that appropriately distributed random matrices can preserve essential information structure regardless of their specific realization. If this hypothesis holds, fixed random projections offer several practical advantages over learned projections below:

*Reduced parameter count*: The projection matrix $W_{\text{in}}$ no longer contributes to the trainable parameter budget, allowing more training resources to be allocated to the processing layers.

*Reduced bandwidth requirement*: Random matrices with known structure (e.g., sparse or circulant patterns) can be generated just-in-time efficiently by custom kernels or custom circuits rather than stored in memory and transferred over the processor-memory interface. This is particularly valuable for architectures like transformers that are often memory-bandwidth limited.

*Reduced memory capacity*: If random matrices are computed on demand, this naturally reduces the memory capacity requirement for both training and inference.

We evaluate this hypothesis empirically in Section 4, comparing the performance of learned versus fixed random input projections across multiple tasks.

### 3.3 MLP SHAPE AND DEPTH STRATEGY

With the wide-narrow-wide MLP paradigm, the total number of MLP parameters for mandatory stages is $d_x d_z + 2L \cdot d_z d_h$. Achieving optimal performance under a total parameter constraint requires one to properly balance the design parameters $d_z$, $d_h$, and $L$.

In general, the higher the latent dimension $d_z$, the more expressive the signal space in which the network solves a task becomes. That expressivity can directly translate into both ease and robustness of learning and performance at convergence. However, this must be counterbalanced by the depth of narrow-wide-narrow MLPs $L$. For many tasks, the deeper the network, the better the performance at convergence. Having a small $d_h$ can indeed enable a larger $L$ seemingly without consequence, but in practice employing an overly deep network can entail certain difficulties.

## 4 EXPERIMENTS AND RESULTS

We evaluate the proposed wide-narrow-wide (Hourglass) MLP architecture against conventional narrow-wide-narrow baselines across multiple generative tasks and datasets. Our experimental design focuses on three key questions: (1) Do Hourglass architectures achieve superior performance-parameter trade-offs compared to conventional designs? (2) How do optimal architectural choices (latent dimension, bottleneck width, and depth) differ between Hourglass and conventional designs? and (3) How does the choice of fixed versus learned input projections affect performance? We conduct systematic architectural searches to characterize the Pareto frontiers for both designs, enabling direct comparison of their efficiency at equivalent parameter budgets.

### 4.1 EXPERIMENTAL SETUP

We evaluate our approach on two image datasets: MNIST (LeCun et al., 2010) and ImageNet-32 (Chrabaszcz et al., 2017), across multiple generative tasks that test different aspects of representation learning and refinement capabilities.

For MNIST, we consider three tasks: (1) generative classification, where the model learns to transform an input image of a digit into a corresponding prototypical image before classification; (2) denoising, where the model removes artificially added Gaussian noise from corrupted images; and (3) super-resolution, where the model upsamples low-resolution inputs to recover high-resolution images.

For ImageNet-32, we focus on the more challenging tasks of (1) denoising natural images with complex textures and structures, and (2) super-resolution that requires preserving fine-grained visual details across diverse object categories.

These tasks are particularly well-suited for evaluating our hypothesis because they require incremental refinement of visual representations — exactly the type of processing we expect to benefit from wider skip connections.

All experiments use the network architecture illustrated in Figure 1(b): an input projection $W_{\text{in}}$, followed by $L$ residual MLP blocks, and an output projection $W_{\text{out}}$. The key difference between the **Hourglass** and **Conventional** models lies in the internal shape of each MLP block. This controlled comparison ensures that both architectures share the same training objectives, input/output configurations, and overall structure, isolating the effect of skip connection placement.

Our architectural search systematically explores the design space defined by: latent dimension $d_z$, hidden dimension $d_h$, and the number of residual blocks $L$. Additionally, we investigate whether the input projection $W_{\text{in}}$ should be learned end-to-end or fixed at random initialization. Detailed experimental settings are provided in Appendix A.3.

### 4.2 MAIN RESULTS AND OBSERVATIONS

We evaluate both architectures by characterizing their performance-parameter Pareto frontiers for each dataset and task combination. The Pareto frontier captures the trade-off between model complexity (number of parameters) and performance (measured by PSNR and SSIM). A model is Pareto-optimal if no other model achieves better performance with fewer parameters—these models represent the most efficient designs at their respective parameter budgets.

Our analysis reveals that Hourglass architectures consistently achieve superior Pareto frontiers compared to conventional designs across all tested tasks. As parameter budgets increase, the optimal Hourglass configurations favor deeper networks with wider latent dimensions but narrower bottleneck dimensions. Additionally, while Hourglass architectures inherently require dimensional expansion for optimal performance, we observe that conventional MLPs can also benefit from random input projections that preserve dimensionality.

### 4.2.1 GENERATIVE CLASSIFICATION TASK

An MNIST generative classification task requires a model to take in an input digit image, generates a prototypical digit image, and then makes a classification based on the latter. Figure 2(b) shows qualitative examples from the Hourglass model. For model training, one image per digit class is chosen to serve as the ground truth digit image.

Figure 2(a) compares the Pareto frontiers of Hourglass and conventional MLPs on the MNIST generative classification task. As shown in Figure 2(a), the Hourglass architecture consistently achieves a better performance–complexity trade-off, reaching higher PSNR values across a wide range of parameter counts. In particular, when the required accuracy is low in the 26 dB range, the Hourglass architecture achieves superior performance with significantly fewer parameters.

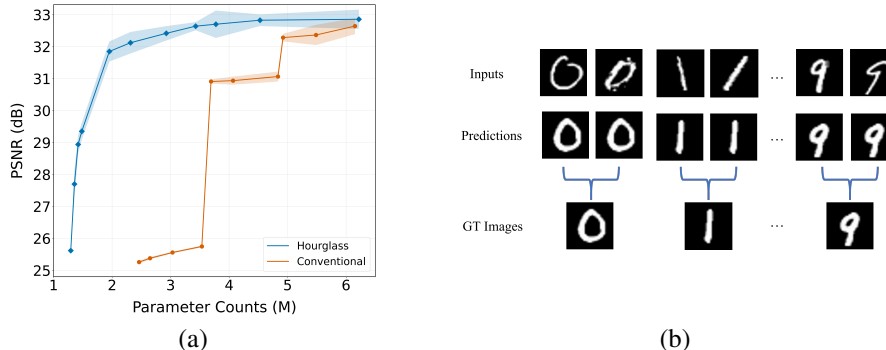

|       |       |
| :---: | :---: |
| (a)   | (b)   |

Figure 2: **Generative Classification Task on MINST.** (a) Performance–complexity Pareto front. Fronts are searched with each configuration repeated 5 times. "Wide–narrow–wide" MLPs outperform conventional "narrow–wide–narrow" ones. (b) Samples predicted by our proposed Hourglass model.

### 4.2.2 GENERATIVE RESTORATION TASKS

We evaluate both architectures on two common generative restoration tasks: denoising and super-resolution. Figures 3 and 4 present the PSNR–parameter Pareto fronts for MNIST and ImageNet-32.

Across datasets and tasks, the proposed wide–narrow–wide (Hourglass) MLP consistently outperforms the conventional narrow–wide–narrow baseline. In denoising (Figure 3(b)), the Hourglass model attains $22.31\,\mathrm{dB}$ PSNR with only $66\mathrm{M}$ parameters, whereas the best conventional model requires $75\mathrm{M}$ to reach the same score. On MNIST (Figure 3(a)), this advantage persists across the entire complexity range.

For super-resolution (Figure 4), the Hourglass design again dominates. On ImageNet-32, it achieves 24.00 dB with 69M parameters, outperforming the 87M-parameter conventional model. The gap is particularly pronounced in the mid-range budget regime. On MNIST, Hourglass MLPs similarly produce better reconstructions at every tested parameter count.

These results suggest that performing residual updates in high-dimensional latent space enhances restoration fidelity and parameter efficiency, especially under tight or mid-range model budgets.

### 4.2.3 PARETO-OPTIMAL ARCHITECTURE CONFIGURATIONS

In both denoising and super-resolution tasks, Tables 1 and 2 summarize the best-performing models on ImageNet-32 under various parameter budgets. Three consistent trends emerge:

- **Hourglass models achieve higher PSNR with fewer parameters.** Across denoising and super-resolution tasks, Hourglass architectures consistently surpass the PSNR of conventional models while using substantially fewer parameters, demonstrating superior efficiency.

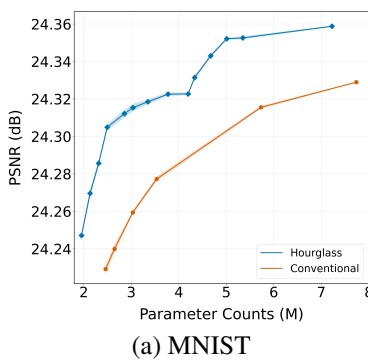 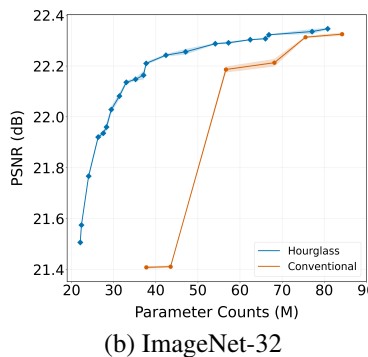

(a) MNIST                    (b) ImageNet-32

Figure 3: **Generative Restoration Task - Denoising.** Performance-complexity Pareto fronts on MINST and ImageNet-32 are searched with each configuration repeated 5 times. Optimal configurations are shown in Table 1.

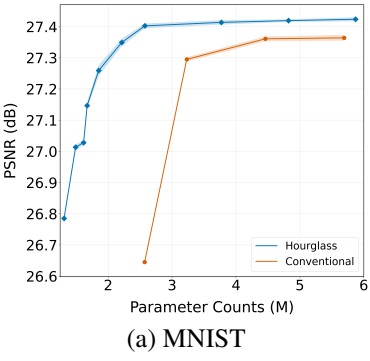 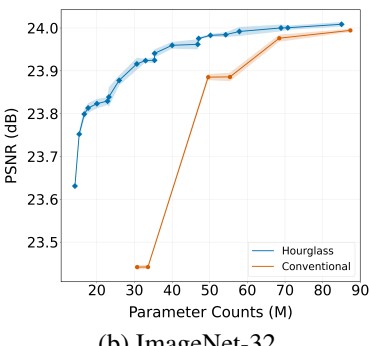

(a) MNIST                    (b) ImageNet-32

Figure 4: **Generative Restoration Task - Super-resolution.** Performance-complexity Pareto fronts on MINST and ImageNet-32 are searched with each configuration repeated 5 times. Optimal configurations are shown in Table 2.

- **Hourglass architectures favor depth and moderate bottlenecks.** Optimal configurations typically use $L = 4$ or 5 with $d_h$ between 270 and 765, in contrast to conventional designs that rely on shallow depth ($L \leqslant 3$) and very wide hidden layers ($d_h \geqslant 3075$).

- **High-dimensional skip connections improve parameter efficiency.** Models with large $d_z$ (commonly 3075 or larger) and relatively small $d_h$ maintain or improve PSNR, confirming the benefits of residual learning in wide latent spaces.

Together, these results confirm that placing skip connections in high-dimensional layers yields more expressive and efficient models with better performance–complexity trade-offs.

### 4.3 Effect of Fixed vs. Trainable Input Projection

To verify our hypothesis in Section 3.2 that randomly initialized projection is sufficient to preserve essential information from input signal, we investigate whether the input projection $W_{in}$ in the Hourglass architecture can be randomly initialized and fixed. On the ImageNet-32 denoising task, we compare two variants under the configuration $(d_z, d_h, L) = (3546, 270, 5)$: (1) *Fixed*: $W_{in}$ is randomly initialized and frozen (20.47M parameters); (2) *Trainable*: $W_{in}$ is updated during training (31.36M parameters).

As shown in Figure 5, the trainable model is only marginally better than the fixed model. These results suggest that the gains from learning $W_{in}$ are minor, and fixed projections offer a strong parameter-efficient alternative—particularly useful in low-resource or hardware-constrained settings.

### 4.4 Ablation Studies on Hourglass MLP Design

To further explore the design trade-offs within the proposed wide–narrow–wide (Hourglass) MLP architecture, we conduct ablation studies focusing on two key hyperparameters: the bottleneck dimension $d_h$ and the number of residual blocks $L$.

| Architecture | Params (M) | $d_z$ | $d_h$ | $L$ | PSNR ($\mu \pm 5\sigma$ dB) |
|---|---|---|---|---|---|
| Conventional | 37.77 | 3072 | 3075 | 1 | $21.408 \pm 0.005$ |
| | 43.52 | 3072 | 4012 | 1 | $21.411 \pm 0.004$ |
| | 56.66 | 3072 | 3075 | 2 | $22.186 \pm 0.012$ |
| | 68.17 | 3072 | 4012 | 2 | $22.213 \pm 0.015$ |
| | 75.55 | 3072 | 3075 | 3 | $22.313 \pm 0.004$ |
| | 84.23 | 3072 | 3546 | 3 | $22.325 \pm 0.007$ |
| Hourglass | 22.07 | 3546 | 8 | 5 | $21.506 \pm 0.007$ |
| | 22.35 | 3546 | 16 | 5 | $21.575 \pm 0.012$ |
| | 24.06 | 3546 | 64 | 5 | $21.767 \pm 0.010$ |
| | 26.33 | 3546 | 128 | 5 | $21.921 \pm 0.010$ |
| | 27.53 | 3546 | 270 | 3 | $21.936 \pm 0.009$ |
| | 28.30 | 3075 | 765 | 2 | $21.960 \pm 0.017$ |
| | 29.45 | 3546 | 270 | 4 | $22.029 \pm 0.012$ |
| | 31.36 | 3546 | 270 | 5 | $22.082 \pm 0.012$ |
| | 33.01 | 3075 | 765 | 3 | $22.136 \pm 0.007$ |
| | 35.19 | 3546 | 270 | 7 | $22.147 \pm 0.005$ |
| | 37.11 | 3546 | 270 | 8 | $22.164 \pm 0.017$ |
| | 37.71 | 3075 | 765 | 4 | $22.210 \pm 0.006$ |
| | 42.42 | 3075 | 765 | 5 | $22.242 \pm 0.005$ |
| | 47.08 | 3075 | 1146 | 4 | $22.256 \pm 0.011$ |
| | 54.13 | 3075 | 1146 | 5 | $22.288 \pm 0.003$ |
| | 57.27 | 3075 | 1560 | 4 | $22.291 \pm 0.005$ |
| | 62.42 | 3546 | 1146 | 5 | $22.303 \pm 0.003$ |
| | 66.04 | 3546 | 1560 | 4 | $22.307 \pm 0.003$ |
| | 66.86 | 3075 | 1560 | 5 | $22.323 \pm 0.002$ |
| | 77.10 | 3546 | 1560 | 5 | $22.335 \pm 0.010$ |
| | 80.82 | 3075 | 2014 | 5 | $22.346 \pm 0.004$ |

Table 1: Pareto optimal model configurations for denoising task on ImageNet-32. An image is linearized to a vector of dimension $d_x = 3072$.

| Architecture | Params (M) | $d_z$ | $d_h$ | $L$ | PSNR ($\mu \pm 5\sigma$ dB) |
|---|---|---|---|---|---|
| Conventional | 30.69 | 3072 | 3075 | 1 | $23.442 \pm 0.005$ |
| | 33.58 | 3072 | 3546 | 1 | $23.442 \pm 0.005$ |
| | 49.58 | 3072 | 3075 | 2 | $23.885 \pm 0.007$ |
| | 55.37 | 3072 | 3546 | 2 | $23.886 \pm 0.010$ |
| | 68.48 | 3072 | 3075 | 3 | $23.976 \pm 0.008$ |
| | 87.37 | 3072 | 3075 | 4 | $23.994 \pm 0.004$ |
| Hourglass | 14.18 | 3546 | 16 | 5 | $23.631 \pm 0.008$ |
| | 15.32 | 3546 | 48 | 5 | $23.752 \pm 0.010$ |
| | 16.67 | 3546 | 86 | 5 | $23.799 \pm 0.012$ |
| | 17.70 | 3546 | 115 | 5 | $23.813 \pm 0.011$ |
| | 20.02 | 4012 | 115 | 5 | $23.823 \pm 0.011$ |
| | 22.83 | 4576 | 115 | 5 | $23.829 \pm 0.009$ |
| | 23.19 | 3546 | 270 | 5 | $23.839 \pm 0.023$ |
| | 25.92 | 3075 | 765 | 3 | $23.878 \pm 0.012$ |
| | 30.63 | 3075 | 765 | 4 | $23.916 \pm 0.014$ |
| | 32.95 | 3075 | 1146 | 3 | $23.923 \pm 0.004$ |
| | 35.32 | 3546 | 765 | 4 | $23.925 \pm 0.007$ |
| | 35.33 | 3075 | 765 | 5 | $23.941 \pm 0.010$ |
| | 40.00 | 3075 | 1146 | 4 | $23.960 \pm 0.008$ |
| | 46.81 | 3546 | 1560 | 3 | $23.962 \pm 0.012$ |
| | 47.05 | 3075 | 1146 | 5 | $23.975 \pm 0.002$ |
| | 50.18 | 3075 | 1560 | 4 | $23.983 \pm 0.004$ |
| | 54.25 | 3546 | 1146 | 5 | $23.984 \pm 0.006$ |
| | 57.87 | 3546 | 1560 | 4 | $23.994 \pm 0.003$ |
| | 68.93 | 3546 | 1560 | 5 | $24.000 \pm 0.002$ |
| | 70.75 | 3546 | 2014 | 4 | $24.001 \pm 0.006$ |
| | 85.03 | 3546 | 2014 | 5 | $24.009 \pm 0.004$ |

Table 2: Pareto optimal model configurations for super-resolution task on ImageNet-32. An image is linearized to a vector of dimension $d_x = 768$.

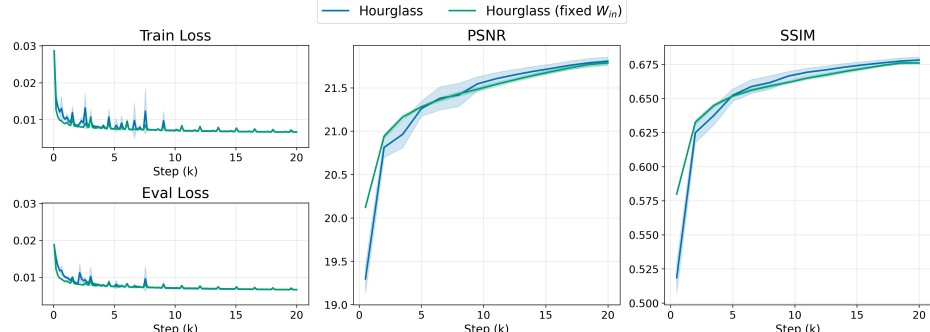

Figure 5: **Input projection fixed with a random projection matrix.** Comparison between fixed and trainable input projection $W_{\text{in}}$ for Hourglass MLP on ImageNet-32 denoising. We use architecture $(d_z, d_h, L) = (3546, 270, 5)$. The fixed-projection model performs comparably to the trainable one.

**Effect of bottleneck width $d_h$:** We fix the high-dimensional residual space to $d_z = 3546$ and the number of residual blocks to $L = 5$, and vary the bottleneck width $d_h$. As shown in Figure 6(a), increasing $d_h$ improves PSNR, but the gains diminish beyond $d_h = 270$. This suggests that moderate bottlenecks are sufficient for high performance, enabling significant parameter savings.

**Effect of residual depth $L$:** We fix $d_z = 3546$, $d_h = 270$, and vary the number of residual blocks $L$. As shown in Figure 6(b), performance improves with deeper stacks, but quickly plateaus around $L = 5$, indicating that relatively shallow Hourglass MLPs are sufficient for strong results.

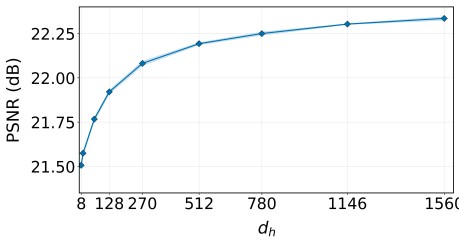

(a) Varying the bottleneck dimension $d_h$

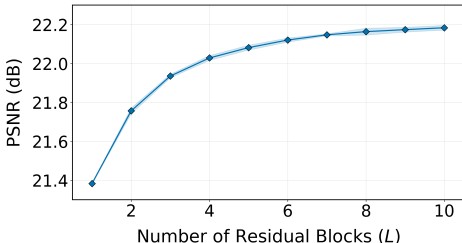

(b) Varying the number of residual blocks $L$

Figure 6: Ablation study of optimal $d_h$ and $L$ dimension for the Hourglass MLP architecture.

## 5 DISCUSSIONS AND FUTURE WORK

Our experimental results demonstrate that wide-narrow-wide (Hourglass) MLP architectures consistently outperform conventional designs across multiple generative tasks, supporting our hypothesis that skip connections at higher dimensions enable more effective incremental refinement. The combination of expanded latent dimensions and random input projections achieves superior performance-parameter trade-offs compared to traditional narrow-wide-narrow architectures.

In this section, we discuss the limitation of our work and the broader implications of "wide-narrow-wide" MLP.

**Scaling to High-Resolution Applications** Due to limited computational capacity, our experiments focus on relatively low-dimensional image datasets to isolate the impact due to architectural differences between conventional and Hourglass MLP designs. However, many real-world applications involve much higher-dimensional inputs—high-resolution images, long sequences, or rich feature representations. Naive MLP approaches become computationally prohibitive for them. We identify two promising directions for scaling our insights to such domains.

First, wide-narrow-wide blocks could be integrated into existing architectures like MLP-Mixer (Tolstikhin et al., 2021) or other similar frameworks. The design of an MLP-Mixer aims at maintaining rich representations while keeping computational costs comparable to MLP designs with a dimensionality equal to the image width modules.

Second, the Hourglass design could enhance U-Net architectures commonly used in image-to-image translation and generative modeling. The input would first be projected into a higher-dimensional latent space before entering the U-Net encoder-decoder pipeline. Then, the concept of wide-narrow-wide shapes can be employed for resolution conversion and for attention.

**Extension to Transformer Architectures.** Looking ahead, the "wide-narrow-wide" MLP architecture presents compelling opportunities for enhancing computational efficiency in modern transformer-based models (Figure 7 (a)). By enabling iterative refinement of representations at expanded dimensionalities, this approach could yield compute-optimal architectures with significantly reduced parameter counts compared to current scaling paradigms.

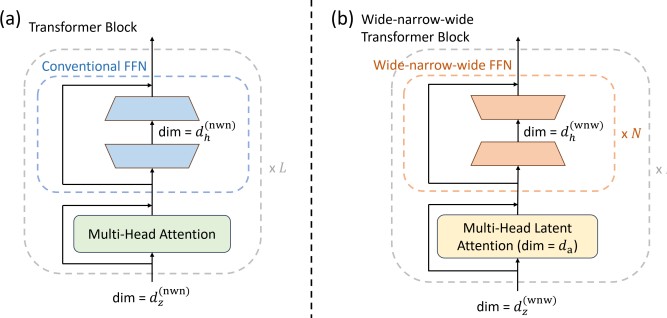

Figure 7: **Extend the wide-narrow-wide intuition to the transformer.** (a) The classic transformer block with Multi-Head Self-Attention and a conventional narrow–wide–narrow FFN. (b) A modified transformer block with block with one or more wide–narrow–wide FFNs and a dimensionality compliant multi-head latent attention sublayer. Components that do not change dimensionality (e.g., normalization, elementwise nonlinearity) are omitted for clarity.

As illustrated in Figure 7 (b), adapting our findings to transformer architectures requires coordinated modifications across self-attention and FF layer. Notably, FF layer cannot operate at expanded dimensions in isolation—the self-attention mechanism must process representations at matching wider dimensionalities to maintain architectural coherence. To preserve computational efficiency, we thus propose incorporating efficient attention mechanisms such as Multi Head Latent Attention (DeepSeek-AI et al., 2025), which maintains reduced attention head sizes while operating over wider representations. Furthermore, our empirical findings on the efficacy of deeper stacks of "wide-narrow-wide" blocks suggest that FF adaptations should incorporate multiple iterative refinement blocks with "wide-narrow-wide" architectural pattern within each FF layer. As a result, such designs could enable more sophisticated representational transformations while maintaining favorable parameter-to-performance ratios, potentially advancing the state-of-the-art in efficient large-scale model architectures.

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

## A  APPENDIX

### A.1  LLM USAGE

In accordance with the official policy on the use of Large Language Models (LLMs), we disclose that LLMs were used in this work solely to aid and polish the writing. Specifically, LLM assistance was employed for grammar checking, improving clarity, and refining the phrasing of certain sentences. No LLM was involved in research ideation, experimental design, data analysis, or the generation of novel scientific content. All technical contributions, results, and interpretations are entirely the work of the authors.

### A.2  REPRODUCIBILITY STATEMENT

We have made every effort to ensure the reproducibility of our results. All source code corresponding to the experiments reported in this paper has been uploaded as part of the supplementary materials, enabling independent verification of our findings. Additional implementation details, hyperparameter settings, and data preprocessing steps are included in the following subsections. Together, these resources allow researchers to fully reproduce the experiments and results presented in this work.

### A.3  DETAILS OF EXPERIMENT SETTINGS

#### A.3.1  SUMMARY OF DATASETS AND TASKS

Table 3 summarizes the datasets, tasks, and input/output signal dimensions.

Table 3: Summary of datasets, tasks, and input/output sizes

| Dataset | Task | Input Size | Output Size | Description |
|---------|------|------------|-------------|-------------|
| MNIST | Generative Classification | $28 \times 28 \times 1$ | $28 \times 28 \times 1$ | Generate GT image for predicted class |
| MNIST | Denoising | $28 \times 28 \times 1$ (noisy) | $28 \times 28 \times 1$ | Remove artificially added noise |
| MNIST | Super-resolution | $14 \times 14 \times 1$ | $28 \times 28 \times 1$ | Recover high-resolution handwritten image |
| ImageNet-32 | Denoising | $32 \times 32 \times 3$ (noisy) | $32 \times 32 \times 3$ | Remove artificially added noise |
| ImageNet-32 | Super-resolution | $16 \times 16 \times 3$ | $32 \times 32 \times 3$ | Recover high-resolution natural scene image |

#### A.3.2  TRAINING SETTING DETAILS

All experiments were conducted using NVIDIA RTX A6000 and RTX 3090 GPUs. The images were mapped to [0,1] before training, and we employed the AdamW (Loshchilov & Hutter, 2017) optimizer with a linear learning rate scheduler and no warm-up period.

**MNIST.** The original training set of 60,000 images was randomly partitioned into 50,000 samples for training and 10,000 for validation, while the original test set of 10,000 images was reserved for final evaluation. The MLP architectural parameters were searched over the ranges $d_h \in [4, 2500]$, $d_z \in [785, 4500]$, and $L \in [1, 40]$, while the learning rate $\in \{1 \times 10^{-4}, 5 \times 10^{-4}, 1 \times 10^{-3}, 5 \times 10^{-3}\}$. . All experiments were repeated 5 times, and we report the mean and standard deviation ($\mu \pm \sigma$) across runs. Note that during grid search, we constrained $d_z > d_x$ and $d_h < d_z$ for the Hourglass architecture, while $d_h > d_z$ for the conventional MLP, following their respective architectural definitions.

- **Generative Classification:** Ground truth images were randomly selected for each digit. Training was conducted with a batch size of 128 for 50 epochs.

- **Denoising:** Noisy images were prepared by adding Gaussian noise (mean = 0, std = 0.25). Training used batch size 128 for 30 epochs.

- **Super-resolution:** Downscaled images were prepared using bicubic interpolation, reducing the original $28 \times 28 \times 1$ images to $14 \times 14 \times 1$. Training applied $4\times$ data augmentation (original, horizontal flip, vertical flip, and combined horizontal-vertical flip) with batch size 128 for 50 epochs.

**ImageNet-32.** The complete original training set of 1,281,167 images was utilized for training, and the original validation set of 50,000 images was randomly split into 25,000 samples for validation and 25,000 for testing. We report the performance on the test set using the model that achieved the lowest validation loss. The MLP architectural parameters were searched over the ranges $d_h \in [4, 2500]$, $d_z \in [8, 2200]$, and $L \in [1, 30]$, while the learning rate $\in \{1 \times 10^{-4}, 3 \times 10^{-4}, 5 \times 10^{-4}, 7 \times 10^{-4}\}$. All experiments were repeated 5 times, and we report the mean and $5\times$ standard deviation ($\mu \pm 5\sigma$) across runs. Note that during grid search, we constrained $d_z > d_x$ and $d_h < d_z$ for the Hourglass architecture, while $d_h > d_z$ for the conventional MLP, following their respective architectural definitions.

- **Denoising:** Noisy images were prepared by adding Gaussian noise (mean = 0, std = 0.25). Training used $4\times$ data augmentation (original, horizontal flip, vertical flip, and combined horizontal-vertical flip) with batch size 512 for 2 epochs.

- **Super-resolution:** Downscaled images were prepared using bicubic interpolation, reducing the original $32 \times 32 \times 3$ images to $16 \times 16 \times 3$. Training applied $4\times$ data augmentation (original, horizontal flip, vertical flip, and combined horizontal-vertical flip) with batch size 512 for 2 epochs.

