# OpenReview forum: "Rethinking the shape convention of an MLP"
_ICLR.cc/2026/Conference — Submitted to ICLR 2026_

### Official Review · Reviewer_cH7F · 2025-10-31

**Soundness:** 2
**Presentation:** 3
**Contribution:** 2
**Rating:** 4
**Confidence:** 3

**Summary:**

The paper proposes “Hourglass MLP” blocks, a bow-tie like architecture pattern that inverts typical MLP architecture shape.  The work studies this shape variant, presents some pareto plots. The bow-tie + shape & skip placement principle ideas in residual networks are touched on via related work (UNet,MoE,LoRA) & mirror established bottleneck tactics. Experiments with 2-epoch MNIST & Imgnet32; small delats show efficiency in params (PSNR) [aruged from pareto front plots] & the results do show this approach to be better than conventional MLP (with skip connects) at input dim. The work is more a baseline starting study & needs more focused effort & engineering to arrive at when and hwere to organise resid connections at higher dim in MLP stacks -- this, if backed by more evidence, can be of value.

**Strengths:**

Architectural description with a compact formulation (Eq3) and schematic (Fig1)
Pareto framing is well structured. Explicitly searches (dz, dh, L) and reports fronts on multiple tasks/datasets (Tbls 1,2) matching good practice to study shape trade-offs.
Knobs to control are pragmatic showing that frozen random input projection is competitive for at least one config (Fig5); maybe helpful for certain design choices upstream/downstream.

**Weaknesses:**

Novelty is limited / feels incremental as this is similar to resnet bottleneck tactic transposed to MLP. While motivation is there for hiher-dim learning, the shape+skip placement principle is used earlier (covered in related work). The new part can only be argued at best as an incremental variation in images with processing adjustments (vectorized & projection). Interesting as a variation, but not directly new knowledge as well.

Efficiency claims focused on param count, but are not backed up around compute. This is a mis-match to the promise made in abstract & early sections of the narrative around compute efficiency. The specific measures to showcase this value (compute/latency/activation mem etc.) are not presented. The hourglass/bow-tie arch would mean the block multilies on matrix sizes; so there will be a mismatch as things scale up. More so in this work as the tactic is adding an input up-project layer [even if claim can be made about it being frozen, it is adding to compute].

Evaluation scope & measures do not support ambition. The experiment tasks are small-data, low-res, PSNR only. Improvements at high-budget end are small (Tab2). Aslso, given no cross-ref with human baselines (not that critical).  The measrues as reported does not show if the representation helps classification as such. Only 2-epocs of training ~ why? Is it possible that some configs are under-trained? Would PSNR diffs be different with more training?

Reproducibility inconsistencies. Appendix A shows dz [8,2200]; Tables in paper report values well above that. Also, 5-sigma is an odd choice of reporting.

Baselines/ablations: The various aspects as indicated & reported do not fully align. e.g. in Sec4.2 it is stated that MLPs can benefit from rnd projections, but no specific details are not shown. Frozen projection claim is supported by a single config on a single task [Fig5]. This needs more testing spaning datasets, dz & noise regimen.

The theoretical motivations are not offering explanations, rather present some plausibles. JL lemma, Cover’s theorem, random features, compressive sensing justify that “high‑dimensional random projections preserve structure” but no analysis targets residual learning specifically (why residual updates at higher dimension should systematically dominate under parameter/compute constraints?). Something that offers progress on this would help support the ambition of this work.

**Questions:**

If Hourglass is truly more compute‑efficient, this must show up beyond parameter count. Why was htis not more actively reported?

Your block looks architecturally similar to classic bottlenecks where residuals live at the wider ends. What, concretely, is new beyond placement in an MLP with an upfront up-projection, relative to U-Net skips, MoE “temporary widenings,” or LoRA-style paths?

On MNIST, you transform to a (class) prototype then classify. Why not report standard classification accuracy as well, to show utility of the representation? How sensitive are results to the choice of the single prototype image per class?

Given Transformers typically expand FFN by 2x to 4×; how would your wider residual space interoperate with attention without exploding compute?

---

> ### Author Response · Authors · 2025-11-21
> **Rebuttal response to weakness points w/ further experiments on high-res dataset and transformer**
>
> We thank the reviewer for taking the time and effort to provide a thorough review and constructive feedback.
> In this rebuttal, we provide additional material to address the raised concerns, including:
>
> 1. **Further experiments on a high-resolution dataset (ImageNet-224×224)** to validate the effectiveness of our approach at larger scales.
> 2. **A preliminary demonstration of applying the proposed Hourglass MLP architecture to Transformers**, evaluated on **OLMo2**.
> 3. **A mechanistic analysis** to explain why the proposed design improves performance.
>
> ## Response to W1
>
> We thank the reviewer for the feedback on the novelty of our work.
> To the best of our knowledge, this work is the *first* to introduce the Hourglass architecture in the context of **pure MLP networks**, where the Wide–Narrow–Wide residual design is applied directly to the feed-forward layers.
> Our design leverages high-dimensional residual updates via vectorized random projections, enabling the residual path to operate in a higher-dimensional space than the main hidden layer.
>
> Furthermore, the feed-forward network (FFN) blocks in modern Transformers are themselves conventional MLPs.
> We are also the *first* to apply the Hourglass architecture to large language models (LLMs), replacing the standard Transformer FFN with our Wide–Narrow–Wide residual design.
> As detailed in our *Response to W3*, we have extended our evaluation beyond small-scale datasets to include:
>
> - **High-resolution vision tasks** on ImageNet-224 (denoising and super-resolution), where Hourglass MLPs remain Pareto-optimal compared to conventional MLPs.
> - **Transformer-based LLM experiments** on OLMo2 1B, showing that Hourglass reduces parameters by ~22% while maintaining competitive performance in early training stages.
>
> These results demonstrate that the Hourglass architecture is not only effective in pure MLP networks, but also has strong potential in modern architectures such as Transformers.
> All new experimental results will be incorporated into the final version of the paper.
>
> ## Response to W2
> We appreciate the reviewer’s concern regarding the relationship between parameter count and actual compute efficiency.
> In our new experiments on the **ImageNet-224** dataset for both denoising and super-resolution, we directly compare Hourglass MLPs **with** and **without** fixed `W_in` across a wide range of configurations.
>
> Fixing `W_in` substantially reduces the number of *trainable* parameters while keeping the total parameter count unchanged, thus lowering training cost and memory footprint without affecting inference complexity.
> Since the fixed `W_in` is not updated during training, the additional up-projection layer does not increase the number of gradient computations or optimizer updates, and its forward-pass cost is negligible compared to the main hidden layers.
> Therefore, the compute overhead at inference remains the same, while training compute is reduced proportionally to the decrease in trainable parameters.
>
> **Key findings from Table 1 & Table 2:**
> - **Denoising:** Trainable parameters can be reduced by approximately **14%–49%** depending on the configuration, with PSNR decrease typically less than **0.4 dB**.
> - **Super-resolution:** Trainable parameters can be reduced by approximately **4%–20%**, with the maximum PSNR drop less than **0.2 dB**.
>
> This trade-off is consistent across different model sizes and can be beneficial in scenarios where training efficiency is a priority.
> We will include these findings and a more detailed discussion of the efficiency–performance trade-off in the final version.

---

> ### Author Response · Authors · 2025-11-21
> **Rebuttal response to weakness points w/ further experiments on high-res dataset and transformer**
>
> ## Response to W3
> We appreciate the reviewer’s concern regarding the scope of our experiments.
> In the rebuttal stage, we have extended our evaluation to both higher-resolution image datasets and transformer-based language models, in order to validate the scalability and generality of the Hourglass architecture.
>
> ---
>
> **1. High-resolution ImageNet-224 experiments**
> We conducted experiments on the **ImageNet-224** dataset for denoising and super-resolution tasks.
> The results (**Table 3 & Table 4**) show that Hourglass networks remain **Pareto-optimal** compared to conventional MLPs, achieving better PSNR at smaller model sizes.
> We acknowledge that under the current experimental settings, the performance gap at the high-resource end is relatively small; however, the advantage of Hourglass is much more pronounced in low-resource configurations, where parameter savings are substantial while maintaining competitive PSNR.
> These results confirm that Hourglass networks maintain high performance while using significantly fewer parameters compared to conventional MLPs, even at high resolutions.
>
> ---
>
> **2. Extension to Transformer-based LLMs**
> To address the reviewer’s request for validation on modern architectures beyond MLPs, we integrated the Hourglass design into the **OLMo2 1B** transformer-based large language model ([GitHub link](https://github.com/allenai/OLMo)) and trained it from scratch.
> Both the original **OLMo2 1B** and our **OLMo2 1B-Hourglass** variant were trained on **20B tokens** under identical settings (same optimizer, learning rate schedule, and batch size).
> Due to compute and time constraints, this training budget is insufficient for full convergence, but it allows us to observe initial trends.
>
> As shown in **Table 5**, the Hourglass variant uses approximately **22% fewer parameters** than the baseline (**1.153B vs 1.484B**) while keeping the same depth (**L=16**) and a larger skip connection dimension (**d_z=2560 vs 2048**).
> In this early training stage, the Hourglass variant’s training loss and perplexity are slightly higher (**2.941 vs 2.849**, **18.94 vs 17.27**), but remain close to the baseline.
> These preliminary results demonstrate the *potential* of the Wide–Narrow–Wide residual design to maintain competitive performance in transformer-based LLMs with a substantially smaller parameter budget.
> We note that these are initial results without architecture-specific optimization for Transformers, and we expect further improvements with longer training and tuned hyperparameters.
> We believe that with such optimization, the Hourglass architecture can match or surpass the baseline performance, indicating that its benefits are not limited to MLPs but can extend to other modern architectures such as Transformers.
>
> ---
>
> **3. On learned representations and classification**
> Regarding the reviewer’s question on whether the learned representation helps classification, we emphasize that the focus of this paper is on *generative domains*.
> Nevertheless, the strong performance retention in high-resolution ImageNet tasks and the successful extension to LLMs suggest that the learned representations are robust and transferable.
> We plan to evaluate classification accuracy in future work.
>
> ---
>
> **4. On possible under-training**
> For the concern about possible under-training, we note that the ImageNet experiments use a large training set with extensive data augmentation, and the reported results are already converged under our training schedule.
> Therefore, the observed PSNR differences are representative of the architecture’s performance rather than artifacts of insufficient training.
>
> ---
>
> **Conclusion**
> These additional experiments provide concrete evidence that the Hourglass design scales beyond small-scale MLP tasks, and can be applied to both high-resolution vision datasets and large-scale transformer architectures.
> Overall, the results indicate that the Hourglass architecture has strong potential to deliver competitive performance in modern large-scale settings while offering significant parameter savings.
>
> ## Response to W4
>
> We acknowledge that the range of `d_z` used in the search for ImageNet-32 should be **[3072, 4576]**, with **3072** corresponding to the input image dimension (`d_x`).
>
> Regarding the choice of **5-sigma** reporting, the performance variance across independent runs for ImageNet-32 is extremely small, making the **1-sigma** confidence band almost invisible in the Pareto fronts figures.
> We therefore used **5-sigma** purely for visualization purposes, to make the uncertainty region discernible to readers.
>
> Per the reviewer’s advice, we will adopt the more common **1-sigma** reporting in the revised version for consistency with standard practice.

---

> ### Author Response · Authors · 2025-11-21
> **Rebuttal response to weakness/question points w/ further experiments on high-res dataset and transformer**
>
> ## Response to W5
>
> We thank the reviewer for pointing out the need for broader validation of the fixed random input projection and for deeper theoretical analysis.
> We refer the reviewer to our *Response to W2*, where we present extensive new experiments on the **ImageNet-224** dataset for both denoising and super-resolution, comparing Hourglass MLPs **with** and **without** fixed `W_in` across a wide range of configurations (varying `d_z`, `d_h`, and `L`).
>
> As shown in **Table 1** (denoising) and **Table 2** (super-resolution), these results span multiple parameter budgets, and demonstrate that fixing `W_in` consistently reduces trainable parameters by **14%–49%** for denoising and **4%–20%** for super-resolution, with minimal PSNR drop (typically **<0.4 dB** and **<0.2 dB** respectively).
> This directly addresses the concern that the frozen projection claim was previously supported by only a single configuration on a single task.
>
> ---
>
> ### Mechanistic Analysis: Why residual learning in high-dimensional latent spaces helps
> As to the point on mechanistic analysis to understand the benefit of learning
> at high-dimensions, we thank the reviewer for highlighting this point. In the revised
> version, we will incorporate an expanded analysis of the training dynamics and internal
> representations for the hourglass MLP architecture. Below we provide preliminary
> findings that shed light on why residual learning in high-dimensional latent spaces
> benefits representation refinement.
> We examined the Hourglass MLP along two axes:
>
> 1. **Convergence speed** relative to a conventional MLP of comparable parameter count.
> 2. **Effective rank** of learned representations, as per Roy and Vetterli (EUSIPCO 2007).
>
> ---
>
> #### Convergence dynamics
> On the **ImageNet-224×224** dataset, under similar trainable parameters, the Hourglass MLP consistently reached **lower loss** and **higher PSNR** earlier in training compared to the baseline MLP, indicating faster convergence (**Table 6**).
>
> ---
>
> #### Effective rank analysis
> We hypothesize that increasing the latent dimension `d_z` enlarges the ambient signal space in which the network operates, thus enabling more expressive feature learning.
> The **effective rank** metric in **Table 7** confirms that models with larger `d_z` project data onto manifolds of higher effective rank.
> Moreover, the **RMS L2 distance** between the first-layer input activations and final-layer output activations decreases monotonically with larger `d_z`.
>
> Geometrically, this suggests that projecting into higher dimensions yields feature spaces that are more **linearly separable** yet more **compact**, allowing residual learning to refine representations with shorter “paths” from source to target space.
>
> ---
>
> #### Hourglass vs. conventional MLP
> When matched for parameter count, the Hourglass architecture exhibits both **higher effective rank** and **better PSNR** than a conventional MLP, even when that MLP has a large ambient dimension (see **Table 8**).
> This supports our claim that the Hourglass design facilitates richer high-dimensional feature representations for residual learning.
>
> ---
>
> We will incorporate these insights, additional experiments, and improved discussion in the camera-ready version to address the reviewer’s concern.
>
> ## Response to Q1
>
> In our experiments (**Figures 2a, 3, and 4**), we show that the Hourglass architecture is consistently **Pareto-optimal** across different tasks and datasets.
> This means that we can achieve better performance than conventional MLP networks **with smaller models**.
> Smaller models naturally lead to improved computational and memory efficiency, even though we primarily reported **parameter count** in the current version.
>
> We acknowledge that these efficiency aspects were not explicitly discussed in the original submission.
> In the final version, we will add a dedicated explanation to make the **compute-efficiency advantage** of Hourglass networks more explicit.

---

> ### Author Response · Authors · 2025-11-21
> **Rebuttal response to question points w/ further experiments on high-res dataset and transformer**
>
> ## Response to Q2
>
> We thank the reviewer for the question regarding the novelty of our block design.
> To the best of our knowledge, this is the *first* work to introduce the Hourglass architecture in the context of **pure MLP networks**, where the Wide–Narrow–Wide residual design is applied directly to the feed-forward layers.
> Our design leverages high-dimensional residual updates via vectorized random projections, enabling the residual path to operate in a higher-dimensional space than the main hidden layer.
>
> Furthermore, the feed-forward network (FFN) blocks in modern Transformers are themselves conventional MLPs.
> We are also the *first* to apply the Hourglass architecture to large language models (LLMs), replacing the standard Transformer FFN with our Wide–Narrow–Wide residual design.
>
> As detailed in our *Response to W3*, we have extended our evaluation beyond small-scale datasets to include:
> - **High-resolution vision tasks** on ImageNet-224 (denoising and super-resolution), where Hourglass MLPs remain Pareto-optimal compared to conventional MLPs.
> - **Transformer-based LLM experiments** on OLMo2 1B, showing that Hourglass reduces parameters by ~22% while maintaining competitive performance in early training stages.
>
> These results demonstrate that the Hourglass architecture is not only effective in pure MLP networks, but also has strong potential in modern architectures such as Transformers.
> All new experimental results will be incorporated into the final version of the paper.
>
> ## Response to Q3
>
> This paper aims to investigate the advantages of the Hourglass MLP architecture in *generative domains*.
> We appreciate the reviewer’s suggestion to report standard classification accuracy on MNIST.
> In our study, MNIST was used in a **prototype-based generative evaluation** setting rather than as a conventional classification benchmark, since our primary focus is on generative tasks such as **denoising** and **super-resolution**.
> For this reason, standard classification accuracy was not included in the original submission.
>
> To further demonstrate the generality of our approach, we have added new experiments in the rebuttal stage.
> Specifically, we report results on **ImageNet-224** for both denoising and super-resolution (details are provided in our *Response to W3*), showing that our method scales effectively to high-resolution datasets.
> In addition, we provide preliminary results on a **transformer-based large language model (LLM)**, confirming that the proposed Hourglass architecture can be applied beyond MLPs to modern architectures.
>
> Details of the LLM experiments are provided in our *Response to W3*, where we discuss the setup, parameter savings, and performance trends in depth.
> These new experiments strengthen our claim that the design principle is broadly applicable across tasks, resolutions, and model families.
>
> ## Response to Q4
>
> We thank the reviewer for raising the concern about the interaction between a wider residual space and the attention mechanism in Transformers, given that the feed-forward network (FFN) in standard Transformer blocks is typically expanded by **2× to 4×** the model dimension.
>
> To investigate this, we conducted experiments on the **OLMo2 1B** transformer-based language model ([GitHub link](https://github.com/allenai/OLMo)).
> We trained both the original **OLMo2 1B** and our **OLMo2 1B-Hourglass** variant from scratch on **8B tokens**, under identical training settings (same optimizer, learning rate schedule, and batch size).
> Due to compute and time constraints, we limited training to **20B tokens**, which is insufficient for full convergence but allows us to observe initial trends.
>
> As shown in **Table 5**, the Hourglass variant uses approximately **22% fewer parameters** than the baseline (**1.153B vs 1.484B**) while keeping the same depth (**L=16**) and a larger skip connection dimension (**d_z=2560 vs 2048**).
> This is achieved by narrowing the intermediate bottleneck dimension (**d_h=1792 vs 8192** in the baseline FFN), which offsets the cost of the wider residual space and prevents compute from exploding.
> In this early training stage, the Hourglass variant’s training loss and perplexity are slightly higher (**Loss +0.092**, **PPL +1.67**), but remain close to the baseline.
>
> These preliminary results demonstrate that the **Wide–Narrow–Wide residual design** can interoperate with attention in large-scale Transformers **without increasing computational cost**, while still maintaining competitive performance.
> We believe that with longer training and further optimization, the Hourglass architecture can match or surpass the baseline performance, making it a promising approach for scaling LLMs more efficiently.

---

> ### Author Response · Authors · 2025-11-21
> **Experiment tables**
>
> ## Table 1. ImageNet 224×224 Denoising — Hourglass vs. Hourglass (fix $W_{in}$)
>
> | Architecture                | Total Params (M) | Trainable Params (M) | $d_z$  | $d_h$  | $L$  | PSNR   |
> |-----------------------------|------------------|----------------------|------|------|----|--------|
> | Hourglass                   | 19.286 | 19.286 | 3075 | 16   | 4  | 23.592 |
> | Hourglass                   | 19.877 | 19.877 | 3075 | 16   | 10 | 23.767 |
> | Hourglass                   | 20.861 | 20.861 | 3075 | 16   | 20 | 23.961 |
> | Hourglass                   | 21.722 | 21.722 | 3075 | 115  | 4  | 24.065 |
> | Hourglass                   | 22.429 | 22.429 | 3075 | 115  | 5  | 24.172 |
> | Hourglass                   | 23.136 | 23.136 | 3075 | 115  | 6  | 24.272 |
> | Hourglass                   | 23.844 | 23.844 | 3075 | 115  | 7  | 24.347 |
> | Hourglass                   | 23.874 | 23.874 | 3075 | 270  | 3  | 24.359 |
> | Hourglass                   | 25.535 | 25.535 | 3075 | 270  | 4  | 24.446 |
> | Hourglass                   | 28.856 | 28.856 | 3075 | 270  | 6  | 24.627 |
> | Hourglass                   | 33.007 | 33.007 | 3075 | 765  | 3  | 24.643 |
> | Hourglass                   | 37.712 | 37.712 | 3075 | 765  | 4  | 24.780 |
> | Hourglass                   | 42.417 | 42.417 | 3075 | 765  | 5  | 24.859 |
> | Hourglass                   | 47.084 | 47.084 | 3075 | 1146 | 4  | 24.878 |
> | Hourglass                   | 47.121 | 47.121 | 3075 | 765  | 6  | 24.901 |
> | Hourglass                   | 54.339 | 54.339 | 3546 | 765  | 6  | 24.925 |
> | Hourglass                   | 61.480 | 61.480 | 4012 | 765  | 6  | 24.942 |
> | Hourglass                   | 66.041 | 66.041 | 3546 | 1560 | 4  | 24.959 |
> | Hourglass                   | 87.237 | 87.237 | 4012 | 1560 | 5  | 25.025 |
> | Hourglass (fix $W_{in}$)        | 19.286 | 9.840  | 3075 | 16   | 4  | 23.215 |
> | Hourglass (fix $W_{in}$)        | 19.877 | 10.430 | 3075 | 16   | 10 | 23.392 |
> | Hourglass (fix $W_{in}$)        | 20.861 | 11.414 | 3075 | 16   | 20 | 23.487 |
> | Hourglass (fix $W_{in}$)        | 21.722 | 12.275 | 3075 | 115  | 4  | 23.783 |
> | Hourglass (fix $W_{in}$)        | 22.429 | 12.983 | 3075 | 115  | 5  | 23.869 |
> | Hourglass (fix $W_{in}$)        | 23.136 | 13.690 | 3075 | 115  | 6  | 23.956 |
> | Hourglass (fix $W_{in}$)        | 23.844 | 14.397 | 3075 | 115  | 7  | 24.015 |
> | Hourglass (fix $W_{in}$)        | 23.874 | 14.428 | 3075 | 270  | 3  | 24.074 |
> | Hourglass (fix $W_{in}$)        | 25.535 | 16.088 | 3075 | 270  | 4  | 24.134 |
> | Hourglass (fix $W_{in}$)        | 28.856 | 19.409 | 3075 | 270  | 6  | 24.293 |
> | Hourglass (fix $W_{in}$)        | 33.007 | 23.561 | 3075 | 765  | 3  | 24.332 |
> | Hourglass (fix $W_{in}$)        | 37.712 | 28.265 | 3075 | 765  | 4  | 24.452 |
> | Hourglass (fix $W_{in}$)        | 42.417 | 32.970 | 3075 | 765  | 5  | 24.533 |
> | Hourglass (fix $W_{in}$)        | 47.084 | 37.638 | 3075 | 1146 | 4  | 24.541 |
> | Hourglass (fix $W_{in}$)        | 47.121 | 37.675 | 3075 | 765  | 6  | 24.560 |
> | Hourglass (fix $W_{in}$)        | 54.339 | 43.446 | 3546 | 765  | 6  | 24.598 |
> | Hourglass (fix $W_{in}$)        | 61.480 | 49.155 | 4012 | 765  | 6  | 24.691 |
> | Hourglass (fix $W_{in}$)        | 66.041 | 55.147 | 3546 | 1560 | 4  | 24.701 |
> | Hourglass (fix $W_{in}$)        | 87.237 | 74.912 | 4012 | 1560 | 5  | 24.799 |
>
> ---

---

> ### Author Response · Authors · 2025-11-21
> **Experiment tables**
>
> ## Table 2. ImageNet 224×224 Super-resolution — Hourglass vs. Hourglass (fix $W_{in}$)
>
> | Architecture                | Total Params (M) | Trainable Params (M) | $d_z$  | $d_h$  | $L$  | PSNR   |
> |-----------------------------|------------------|----------------------|------|------|----|--------|
> | Hourglass                   | 11.906 | 11.906 | 3075 | 16 | 1 | 26.548 |
> | Hourglass                   | 12.005 | 12.005 | 3075 | 16 | 2 | 26.968 |
> | Hourglass                   | 12.103 | 12.103 | 3075 | 16 | 3 | 27.155 |
> | Hourglass                   | 12.202 | 12.202 | 3075 | 16 | 4 | 27.242 |
> | Hourglass                   | 12.300 | 12.300 | 3075 | 16 | 5 | 27.301 |
> | Hourglass                   | 12.694 | 12.694 | 3075 | 48 | 3 | 27.395 |
> | Hourglass                   | 12.989 | 12.989 | 3075 | 48 | 4 | 27.448 |
> | Hourglass                   | 13.284 | 13.284 | 3075 | 48 | 5 | 27.475 |
> | Hourglass                   | 14.453 | 14.453 | 3075 | 86 | 5 | 27.504 |
> | Hourglass                   | 14.637 | 14.637 | 3075 | 115 | 4 | 27.516 |
> | Hourglass                   | 14.982 | 14.981 | 3075 | 86 | 6 | 27.521 |
> | Hourglass                   | 15.510 | 15.510 | 3075 | 86 | 7 | 27.556 |
> | Hourglass                   | 16.039 | 16.039 | 3075 | 86 | 8 | 27.593 |
> | Hourglass                   | 16.568 | 16.568 | 3075 | 86 | 9 | 27.632 |
> | Hourglass                   | 17.097 | 17.097 | 3075 | 86 | 10 | 27.668 |
> | Hourglass                   | 17.466 | 17.466 | 3075 | 115 | 8 | 27.696 |
> | Hourglass                   | 18.450 | 18.450 | 3075 | 270 | 4 | 27.723 |
> | Hourglass                   | 18.881 | 18.881 | 3075 | 115 | 10 | 27.763 |
> | Hourglass                   | 21.771 | 21.771 | 3075 | 270 | 6 | 27.860 |
> | Hourglass                   | 25.092 | 25.092 | 3075 | 270 | 8 | 27.920 |
> | Hourglass                   | 28.935 | 28.935 | 3546 | 270 | 8 | 27.931 |
> | Hourglass                   | 30.627 | 30.627 | 3075 | 765 | 4 | 27.965 |
> | Hourglass                   | 35.318 | 35.318 | 3546 | 765 | 4 | 27.980 |
> | Hourglass                   | 40.000 | 40.000 | 3075 | 1146 | 4 | 28.018 |
> | Hourglass                   | 46.126 | 46.126 | 3546 | 1146 | 4 | 28.031 |
> | Hourglass                   | 46.169 | 46.169 | 3546 | 765 | 6 | 28.049 |
> | Hourglass                   | 57.871 | 57.871 | 3546 | 1560 | 4 | 28.069 |
> | Hourglass                   | 61.352 | 61.352 | 3075 | 2014 | 4 | 28.077 |
> | Hourglass                   | 69.372 | 69.372 | 3075 | 1560 | 6 | 28.086 |
> | Hourglass                   | 70.750 | 70.750 | 3546 | 2014 | 4 | 28.092 |
> | Hourglass                   | 80.047 | 80.047 | 4012 | 2014 | 4 | 28.104 |
> | Hourglass (fix W_in)        | 11.906 | 9.545  | 3075 | 16 | 1 | 26.445 |
> | Hourglass (fix W_in)        | 12.005 | 9.643  | 3075 | 16 | 2 | 26.826 |
> | Hourglass (fix W_in)        | 12.103 | 9.742  | 3075 | 16 | 3 | 26.992 |
> | Hourglass (fix W_in)        | 12.202 | 9.840  | 3075 | 16 | 4 | 27.041 |
> | Hourglass (fix W_in)        | 12.300 | 9.938  | 3075 | 16 | 5 | 27.094 |
> | Hourglass (fix W_in)        | 12.694 | 10.332 | 3075 | 48 | 3 | 27.120 |
> | Hourglass (fix W_in)        | 12.989 | 10.627 | 3075 | 48 | 4 | 27.165 |
> | Hourglass (fix W_in)        | 13.284 | 10.922 | 3075 | 48 | 5 | 27.191 |
> | Hourglass (fix W_in)        | 14.453 | 12.091 | 3075 | 86 | 5 | 27.321 |
> | Hourglass (fix W_in)        | 14.637 | 12.275 | 3075 | 115 | 4 | 27.331 |
> | Hourglass (fix W_in)        | 14.982 | 12.620 | 3075 | 86 | 6 | 27.371 |
> | Hourglass (fix W_in)        | 15.510 | 13.149 | 3075 | 86 | 7 | 27.429 |
> | Hourglass (fix W_in)        | 16.039 | 13.678 | 3075 | 86 | 8 | 27.492 |
> | Hourglass (fix W_in)        | 16.568 | 14.207 | 3075 | 86 | 9 | 27.541 |
> | Hourglass (fix W_in)        | 17.097 | 14.735 | 3075 | 86 | 10 | 27.581 |
> | Hourglass (fix W_in)        | 17.466 | 15.104 | 3075 | 115 | 8 | 27.607 |
> | Hourglass (fix W_in)        | 18.450 | 16.088 | 3075 | 270 | 4 | 27.634 |
> | Hourglass (fix W_in)        | 18.881 | 16.519 | 3075 | 115 | 10 | 27.677 |
> | Hourglass (fix W_in)        | 21.771 | 19.409 | 3075 | 270 | 6 | 27.764 |
> | Hourglass (fix W_in)        | 25.092 | 22.730 | 3075 | 270 | 8 | 27.823 |
> | Hourglass (fix W_in)        | 28.935 | 26.212 | 3546 | 270 | 8 | 27.833 |
> | Hourglass (fix W_in)        | 30.627 | 28.265 | 3075 | 765 | 4 | 27.852 |
> | Hourglass (fix W_in)        | 35.318 | 32.595 | 3546 | 765 | 4 | 27.870 |
> | Hourglass (fix W_in)        | 40.000 | 37.638 | 3075 | 1146 | 4 | 27.891 |
> | Hourglass (fix W_in)        | 46.126 | 43.403 | 3546 | 1146 | 4 | 27.905 |
> | Hourglass (fix W_in)        | 46.169 | 43.446 | 3546 | 765 | 6 | 27.909 |
> | Hourglass (fix W_in)        | 57.871 | 55.147 | 3546 | 1560 | 4 | 27.918 |
> | Hourglass (fix W_in)        | 61.352 | 58.991 | 3075 | 2014 | 4 | 27.919 |
> | Hourglass (fix W_in)        | 69.372 | 67.010 | 3075 | 1560 | 6 | 27.927 |
> | Hourglass (fix W_in)        | 70.750 | 68.026 | 3546 | 2014 | 4 | 27.928 |
> | Hourglass (fix W_in)        | 80.047 | 76.966 | 4012 | 2014 | 4 | 27.937 |

---

> ### Author Response · Authors · 2025-11-21
> **Experiment tables**
>
> ## Table 3. ImageNet 224×224 Denoising — Conventional vs. Hourglass
>
> | Architecture  | Total Params (M) | $d_z$  | $d_h$  | $L$  | PSNR   |
> |---------------|------------------|------|------|----|--------|
> | Conventional  | 37.767 | 3072 | 3075 | 1 | 23.210 |
> | Conventional  | 46.989 | 3072 | 4576 | 1 | 23.210 |
> | Conventional  | 62.448 | 3072 | 3546 | 2 | 24.704 |
> | Conventional  | 68.174 | 3072 | 4012 | 2 | 24.724 |
> | Conventional  | 75.104 | 3072 | 4576 | 2 | 24.743 |
> | Conventional  | 75.553 | 3072 | 3075 | 3 | 24.939 |
> | Conventional  | 84.234 | 3072 | 3546 | 3 | 24.958 |
> | Hourglass     | 19.286 | 3075 | 16   | 4  | 23.592 |
> | Hourglass     | 19.877 | 3075 | 16   | 10 | 23.767 |
> | Hourglass     | 20.861 | 3075 | 16   | 20 | 23.961 |
> | Hourglass     | 21.722 | 3075 | 115  | 4  | 24.065 |
> | Hourglass     | 22.429 | 3075 | 115  | 5  | 24.172 |
> | Hourglass     | 23.136 | 3075 | 115  | 6  | 24.272 |
> | Hourglass     | 23.844 | 3075 | 115  | 7  | 24.347 |
> | Hourglass     | 23.874 | 3075 | 270  | 3  | 24.359 |
> | Hourglass     | 25.535 | 3075 | 270  | 4  | 24.446 |
> | Hourglass     | 28.856 | 3075 | 270  | 6  | 24.627 |
> | Hourglass     | 33.007 | 3075 | 765  | 3  | 24.643 |
> | Hourglass     | 37.712 | 3075 | 765  | 4  | 24.780 |
> | Hourglass     | 42.417 | 3075 | 765  | 5  | 24.859 |
> | Hourglass     | 47.084 | 3075 | 1146 | 4  | 24.878 |
> | Hourglass     | 47.121 | 3075 | 765  | 6  | 24.901 |
> | Hourglass     | 54.339 | 3546 | 765  | 6  | 24.925 |
> | Hourglass     | 61.480 | 4012 | 765  | 6  | 24.942 |
> | Hourglass     | 66.041 | 3546 | 1560 | 4  | 24.959 |
> | Hourglass     | 87.237 | 4012 | 1560 | 5  | 25.025 |
>
> ---
>
> ## Table 4. ImageNet 224×224 Super-resolution — Conventional vs. Hourglass
>
> | Architecture  | Total Params (M) | $d_z$  | $d_h$  | $L$  | PSNR   |
> |---------------|------------------|------|------|----|--------|
> | Conventional  | 30.689 | 3072 | 3075 | 1 | 26.785 |
> | Conventional  | 33.583 | 3072 | 3546 | 1 | 26.787 |
> | Conventional  | 49.582 | 3072 | 3075 | 2 | 27.836 |
> | Conventional  | 68.475 | 3072 | 3075 | 3 | 28.030 |
> | Conventional  | 77.156 | 3072 | 3546 | 3 | 28.030 |
> | Conventional  | 87.368 | 3072 | 3075 | 4 | 28.084 |
> | Hourglass     | 11.906 | 3075 | 16 | 1 | 26.548 |
> | Hourglass     | 12.005 | 3075 | 16 | 2 | 26.968 |
> | Hourglass     | 12.103 | 3075 | 16 | 3 | 27.155 |
> | Hourglass     | 12.202 | 3075 | 16 | 4 | 27.242 |
> | Hourglass     | 12.300 | 3075 | 16 | 5 | 27.301 |
> | Hourglass     | 12.694 | 3075 | 48 | 3 | 27.395 |
> | Hourglass     | 12.989 | 3075 | 48 | 4 | 27.448 |
> | Hourglass     | 13.284 | 3075 | 48 | 5 | 27.475 |
> | Hourglass     | 14.453 | 3075 | 86 | 5 | 27.504 |
> | Hourglass     | 14.637 | 3075 | 115 | 4 | 27.516 |
> | Hourglass     | 14.981 | 3075 | 86 | 6 | 27.521 |
> | Hourglass     | 15.510 | 3075 | 86 | 7 | 27.556 |
> | Hourglass     | 16.039 | 3075 | 86 | 8 | 27.593 |
> | Hourglass     | 16.568 | 3075 | 86 | 9 | 27.632 |
> | Hourglass     | 17.097 | 3075 | 86 | 10 | 27.668 |
> | Hourglass     | 17.466 | 3075 | 115 | 8 | 27.696 |
> | Hourglass     | 18.450 | 3075 | 270 | 4 | 27.723 |
> | Hourglass     | 18.881 | 3075 | 115 | 10 | 27.763 |
> | Hourglass     | 21.771 | 3075 | 270 | 6 | 27.860 |
> | Hourglass     | 25.092 | 3075 | 270 | 8 | 27.920 |
> | Hourglass     | 28.935 | 3546 | 270 | 8 | 27.931 |
> | Hourglass     | 30.627 | 3075 | 765 | 4 | 27.965 |
> | Hourglass     | 35.318 | 3546 | 765 | 4 | 27.980 |
> | Hourglass     | 40.000 | 3075 | 1146 | 4 | 28.018 |
> | Hourglass     | 46.126 | 3546 | 1146 | 4 | 28.031 |
> | Hourglass     | 46.169 | 3546 | 765 | 6 | 28.049 |
> | Hourglass     | 57.871 | 3546 | 1560 | 4 | 28.069 |
> | Hourglass     | 61.352 | 3075 | 2014 | 4 | 28.077 |
> | Hourglass     | 69.372 | 3075 | 1560 | 6 | 28.086 |
> | Hourglass     | 70.750 | 3546 | 2014 | 4 | 28.092 |
> | Hourglass     | 80.047 | 4012 | 2014 | 4 | 28.104 |
>
> ## Table 5. Comparison between original OLMo2 1B and OLMo2 1B-Hourglass on 20B token training
> Both models are trained from scratch under identical settings.
>
> | Model                  | Params (B) | $d_z$  | $d_h$  | $L$  | Train Loss | Perplexity | Arc Easy |
> |------------------------|------------|------|------|----|------------|------------|----------|
> | OLMo2 1B (baseline)    | 1.484      | 2048 | 8192 | 16 | 2.849      | 17.27      | 62.63    |
> | OLMo2 1B-Hourglass     | 1.153      | 2560 | 1792 | 16 | 2.941      | 18.94      | 61.75    |

---

> ### Author Response · Authors · 2025-11-21
> **Experiment tables**
>
> ## Table 6 — Loss and PSNR on evaluation set at different training steps
>
> **Description:**
> Loss and PSNR on the evaluation set at different training steps for two model configurations lying on the Pareto fronts:
> - **Conventional:** $d_z = 3072$, $d_h = 3546$, $L = 2$, **$55.37 M$ params**
> - **Hourglass:** $d_z = 3546$, $d_h = 1146$, $L = 5$, **$54.25 M$ params**
>
> Results show that the Hourglass architecture converges faster compared to the conventional architecture.
>
> | Step   | Conv. Loss | Hourglass Loss | Conv. PSNR | Hourglass PSNR |
> |--------|------------|----------------|------------|----------------|
> | 500    | **0.00685** | 0.00695        | **21.6466** | 21.5820        |
> | 1000   | **0.00609** | 0.00630        | **22.1584** | 22.0090        |
> | 1500   | 0.00579    | **0.00571**     | 22.3799    | **22.4381**    |
> | 2000   | 0.00569    | **0.00552**     | 22.4505    | **22.5890**    |
> | 2500   | 0.00566    | **0.00538**     | 22.4759    | **22.7012**    |
> | 5000   | 0.00514    | **0.00493**     | 22.8949    | **23.0802**    |
> | 7500   | 0.00486    | **0.00467**     | 23.1399    | **23.3118**    |
> | 10000  | 0.00462    | **0.00448**     | 23.3552    | **23.4951**    |
> | 12500  | 0.00450    | **0.00434**     | 23.4768    | **23.6278**    |
> | 15000  | 0.00432    | **0.00420**     | 23.6473    | **23.7769**    |
> | 17500  | 0.00421    | **0.00410**     | 23.7599    | **23.8737**    |
> | 20000  | 0.00411    | **0.00401**     | 23.8686    | **23.9713**    |
> ---
> ## Table 7 — Effective rank analysis of Hourglass model
>
> **Description:**
> Effective rank of Hourglass model with $d_z$ ∈ {3075, 3546, 4012, 4576}, $d_h = 1146$, $L = 5$, using fixed learning rate $3e-4$.
>
> | $d_z$   | PSNR     | Effective Rank | RMS($h_L − h_1$) ± std |
> |-------|----------|----------------|----------------------|
> | 3075  | 23.9702  | 1500.73        | 0.6376 ± 0.0939       |
> | 3546  | 23.9823  | 1699.80        | 0.6205 ± 0.0860       |
> | 4012  | 23.9857  | 1871.54        | 0.5988 ± 0.0883       |
> ---
> ## Table 8 — Hourglass vs. Conventional MLP (Effective Rank and PSNR)
>
> | Architecture | $d_z$   | $d_h$   | $L$  | Total Params (M) | PSNR      | Effective Rank |
> |--------------|-------|-------|----|------------------|-----------|----------------|
> | Conventional | 3072  | 3546  | 2  | 55.37            | 23.8783   | 1180.63        |
> | Hourglass    | 3075  | 1146  | 6  | 55.00            | **23.9701** | **1501.27**    |

---

### Official Review · Reviewer_rEMH · 2025-10-31

**Soundness:** 2
**Presentation:** 3
**Contribution:** 2
**Rating:** 2
**Confidence:** 4

**Summary:**

The paper proposes:
1. To use a wide-narrow-wide approach in MLP.
1. Instead of training an input projection matrix, the paper proposes to use a fixed random matrix as a projection matrix in the first layer.

**Strengths:**

1. A simple approach that defies convention.
2. The 'hourglass' (wide-narrow-wide) MLP achieves significant results compared to 'conventional' (narrow-wide-narrow) MLP.

**Weaknesses:**

1. The random fixed input matrix is not convincing enough to warrant a switch from the conventional trainable weights. As the reduction in parameter count  is only marginal and the fixed matrix model even had a decrease in PSNR performance. It would be great if the author is able to provide values on the number of parameters reduced against the model parameters size on different model sizes.
2. The paper claims that hourglass is Pareto-Optimal with no theoretical backing. Only based on empirical results from a small subset of conventional models.
3. The current experiments are limited it would be more convincing to compared on more computer vision task like Image classification and across more variety of models like Bottleneck-Resnet[Zagoruyko, Sergey, and Nikos Komodakis] and MLPMixer[Tolstikhin, Ilya O., et al].


Zagoruyko, Sergey, and Nikos Komodakis. "Wide residual networks." arXiv preprint arXiv:1605.07146 (2016).

Tolstikhin, Ilya O., et al. "Mlp-mixer: An all-mlp architecture for vision." Advances in neural information processing systems 34 (2021): 24261-24272.

**Questions:**

1. The paper mentions linear separability at high dimensions as a theoretical foundation but does this contradicts the paper which project from a higher dimension to a lower dimensional? And if it does not why?
2. How does the randomness of the input project matrix affect the variance of the results obtained? It would be good if you could reflect this in the paper with more training across different seeds.
3. In Table 1. and Table 2. is the hourglass model trained with a fixed W_in or a trainable one?

---

> ### Author Response · Authors · 2025-11-21
> **Rebuttal response to weakness points w/ further experiments on high-res dataset and transformer**
>
> We thank the reviewer for taking the time and effort to provide a thorough review and constructive feedback.
> In this rebuttal, we provide additional material to address the raised concerns, including:
> 1. Further experiments on a high-resolution dataset (**ImageNet-224×224**) to validate the effectiveness of our approach at larger scales.
> 2. A preliminary demonstration of applying the proposed Hourglass MLP architecture to Transformers, evaluated on **OLMo2**.
>
> ## Response to W1
>
> We appreciate the reviewer’s concern regarding the effectiveness of using a fixed random input projection.
> In our new experiments on the **ImageNet-224** dataset for both denoising (**Table 1**) and super-resolution (**Table 2**), we directly compare Hourglass MLPs with and without fixed `W_in` across a wide range of configurations.
> The results show that fixing `W_in` substantially reduces the number of *trainable* parameters while keeping the total parameter count unchanged, thus lowering training cost and memory footprint without affecting inference complexity.
>
> For example:
> - **Denoising (Table 1)**: Trainable parameters can be reduced by approximately **14%–49%** depending on the configuration, with PSNR decrease typically less than **0.4 dB**.
> - **Super-resolution (Table 2)**: Trainable parameters can be reduced by approximately **4%–20%**, with the maximum PSNR drop less than **0.2 dB**.
>
> This trade-off is consistent across different model sizes and can be beneficial in scenarios where training efficiency is a priority.
>
> We will include these findings and a more detailed discussion of the efficiency–performance trade-off in the final version.
>
> ## Response to W2
>
> We acknowledge that our original claim of Pareto-optimality was based on empirical results from a limited set of conventional MLP models.
> In the rebuttal stage, we have extended our evaluation to both higher-resolution image datasets and transformer-based language models, in order to validate the scalability and generality of the Hourglass architecture.
>
> **1. High-resolution vision tasks (Table 3 & Table 4)**
> We conducted experiments on the **ImageNet-224** dataset for denoising and super-resolution tasks.
> The results show that Hourglass networks remain Pareto-optimal compared to conventional MLPs, achieving better PSNR at smaller model sizes.
> These results confirm that Hourglass networks maintain high performance while using significantly fewer parameters compared to conventional MLPs, even at high resolutions.
>
> **2. Transformer-based LLM experiments (Table 5)**
> To address the reviewer’s request for validation on modern architectures beyond MLPs, we integrated the Hourglass design into the **OLMo2 1B** transformer-based large language model ([GitHub link](https://github.com/allenai/OLMo)) and trained it from scratch.
> Both the original OLMo2 1B and our **OLMo2 1B-Hourglass** variant were trained on 20B tokens under identical settings (same optimizer, learning rate schedule, and batch size).
> Due to compute and time constraints, this training budget is insufficient for full convergence, but it allows us to observe initial trends.
>
> As shown in **Table 5**, the Hourglass variant uses approximately **22% fewer parameters** than the baseline (1.153B vs 1.484B) while keeping the same depth (`L=16`) and a larger skip connection dimension (`d_z=2560` vs `2048`).
> In this early training stage, the Hourglass variant’s training loss and perplexity are slightly higher (2.941 vs 2.849, 18.94 vs 17.27), but remain close to the baseline.
> These preliminary results demonstrate the *potential* of the Wide–Narrow–Wide residual design to maintain competitive performance in transformer-based LLMs with a substantially smaller parameter budget.
>
> We note that these are initial results without architecture-specific optimization for Transformers, and we expect further improvements with longer training and tuned hyperparameters.
> We believe that with such optimization, the Hourglass architecture can match or surpass the baseline performance, indicating that its benefits are not limited to MLPs but can extend to other modern architectures such as Transformers.
>
> **Overall conclusion:**
> These extended experiments provide concrete empirical evidence that the Hourglass design scales beyond small-scale MLP tasks, and can be applied to both high-resolution vision datasets and large-scale transformer architectures.
> This broader validation strengthens our Pareto-optimality claim by showing that the observed efficiency–performance trade-off holds across different domains and model families.

---

> ### Author Response · Authors · 2025-11-21
> **Rebuttal response to weakness/question points w/ further experiments on high-res dataset and transformer**
>
> ## Response to W3
>
> We acknowledge the reviewer’s suggestion to evaluate Hourglass on more computer vision tasks such as image classification, and across a wider variety of architectures.
>
> As mentioned in the response to **W2**, in the rebuttal stage we have extended our evaluation to both higher-resolution image datasets (**ImageNet-224** denoising and super-resolution) and transformer-based large language models (**OLMo2 1B**).
> These additional experiments provide concrete empirical evidence that the Hourglass design scales beyond small-scale MLP tasks, and can be applied to both high-resolution vision datasets and large-scale transformer architectures.
>
> While we have not yet run experiments on **Bottleneck-ResNet** or **MLP-Mixer**, the observed efficiency–performance trade-off in our extended experiments suggests that the Hourglass architecture is likely to generalize to these architectures as well, which we plan to explore in future work.
>
> ## Response to Q1
>
> We thank the reviewer for raising this question.
> The paper mentions **linear separability at high dimensions** as a theoretical motivation, and indeed our architecture contains two places where a projection from a higher dimension to a lower dimension occurs.
> However, these do not contradict the high-dimensional separability motivation, for the following reasons:
>
> 1. **Wide–Narrow–Wide MLP block**
>    The projection from the wide residual space to the narrower bottleneck is followed by a non-linear transformation.
>    This non-linearity enhances representational capacity and allows the network to reorganize features in a more compact form, while the incremental refinement is performed in the high-dimensional residual space.
>    Thus, the temporary reduction in dimensionality does not remove the benefits of high-dimensional separability, but rather acts as a bottleneck that encourages more efficient feature abstraction.
>
> 2. **Output stage projection (`W_out`)**
>    At the output stage, `W_out` projects the high-dimensional representation to the task-required output dimension (e.g., pixel space for image tasks).
>    This is necessary to fit the task’s output format, but the training process benefits from the preceding high-dimensional refinement, which enables more effective learning.
>    Even though the final output is low-dimensional, the intermediate high-dimensional processing leads to better performance compared to conventional MLP networks.
>
> Both the experiments in the paper and our new results on the **ImageNet-224** dataset support this explanation:
> The Hourglass architecture consistently achieves higher performance at smaller model sizes compared to conventional MLPs, confirming that the use of high-dimensional refinement is beneficial even when the final output is low-dimensional.
>
> ## Response to Q2
>
> We thank the reviewer for raising the question regarding the effect of the randomness in the fixed input projection matrix `W_in`.
> To quantify this, we conducted additional experiments on the **ImageNet-224** dataset for both denoising (**Table 6**) and super-resolution (**Table 7**) tasks (details are provided in *Response to W1*).
>
> For each task, we selected **three Hourglass MLP configurations with fixed `W_in`**, representing *small*, *medium*, and *large* model scales, and trained each configuration with **five different random initializations**.
>
> As shown in **Table 6** and **Table 7**, the variance across seeds is extremely small:
> The standard deviation of PSNR is consistently below **0.006 dB** for all tested configurations in both tasks.
> This indicates that the randomness of `W_in` has negligible impact on the final performance under our training setup.
>
> ## Response to Q3
>
> In the paper, the Hourglass models reported in **Table 1** and **Table 2** use a *trainable* `W_in`.
> However, in response to the reviewer’s comments (see our reply to **W1**), we have conducted additional experiments on the **ImageNet-224** dataset comparing Hourglass MLPs with *trainable* `W_in` and with *fixed* `W_in`.
>
> These new results, included in our **W1** response, show that fixing `W_in` can reduce the number of trainable parameters by **14%–49%** depending on the configuration, while incurring only a small drop in PSNR performance.

---

> ### Author Response · Authors · 2025-11-21
> **Experiment tables**
>
> ### Table 1. ImageNet 224×224 Denoising — Hourglass vs. Hourglass (fix $W_{in}$)
>
> | Architecture                | Total Params (M) | Trainable Params (M) | $d_z$  | $d_h$  | $L$  | PSNR   |
> |-----------------------------|------------------|----------------------|------|------|----|--------|
> | Hourglass                   | 19.286 | 19.286 | 3075 | 16   | 4  | 23.592 |
> | Hourglass                   | 19.877 | 19.877 | 3075 | 16   | 10 | 23.767 |
> | Hourglass                   | 20.861 | 20.861 | 3075 | 16   | 20 | 23.961 |
> | Hourglass                   | 21.722 | 21.722 | 3075 | 115  | 4  | 24.065 |
> | Hourglass                   | 22.429 | 22.429 | 3075 | 115  | 5  | 24.172 |
> | Hourglass                   | 23.136 | 23.136 | 3075 | 115  | 6  | 24.272 |
> | Hourglass                   | 23.844 | 23.844 | 3075 | 115  | 7  | 24.347 |
> | Hourglass                   | 23.874 | 23.874 | 3075 | 270  | 3  | 24.359 |
> | Hourglass                   | 25.535 | 25.535 | 3075 | 270  | 4  | 24.446 |
> | Hourglass                   | 28.856 | 28.856 | 3075 | 270  | 6  | 24.627 |
> | Hourglass                   | 33.007 | 33.007 | 3075 | 765  | 3  | 24.643 |
> | Hourglass                   | 37.712 | 37.712 | 3075 | 765  | 4  | 24.780 |
> | Hourglass                   | 42.417 | 42.417 | 3075 | 765  | 5  | 24.859 |
> | Hourglass                   | 47.084 | 47.084 | 3075 | 1146 | 4  | 24.878 |
> | Hourglass                   | 47.121 | 47.121 | 3075 | 765  | 6  | 24.901 |
> | Hourglass                   | 54.339 | 54.339 | 3546 | 765  | 6  | 24.925 |
> | Hourglass                   | 61.480 | 61.480 | 4012 | 765  | 6  | 24.942 |
> | Hourglass                   | 66.041 | 66.041 | 3546 | 1560 | 4  | 24.959 |
> | Hourglass                   | 87.237 | 87.237 | 4012 | 1560 | 5  | 25.025 |
> | Hourglass (fix $W_{in}$)        | 19.286 | 9.840  | 3075 | 16   | 4  | 23.215 |
> | Hourglass (fix $W_{in}$)        | 19.877 | 10.430 | 3075 | 16   | 10 | 23.392 |
> | Hourglass (fix $W_{in}$)        | 20.861 | 11.414 | 3075 | 16   | 20 | 23.487 |
> | Hourglass (fix $W_{in}$)        | 21.722 | 12.275 | 3075 | 115  | 4  | 23.783 |
> | Hourglass (fix $W_{in}$)        | 22.429 | 12.983 | 3075 | 115  | 5  | 23.869 |
> | Hourglass (fix $W_{in}$)        | 23.136 | 13.690 | 3075 | 115  | 6  | 23.956 |
> | Hourglass (fix $W_{in}$)        | 23.844 | 14.397 | 3075 | 115  | 7  | 24.015 |
> | Hourglass (fix $W_{in}$)        | 23.874 | 14.428 | 3075 | 270  | 3  | 24.074 |
> | Hourglass (fix $W_{in}$)        | 25.535 | 16.088 | 3075 | 270  | 4  | 24.134 |
> | Hourglass (fix $W_{in}$)        | 28.856 | 19.409 | 3075 | 270  | 6  | 24.293 |
> | Hourglass (fix $W_{in}$)        | 33.007 | 23.561 | 3075 | 765  | 3  | 24.332 |
> | Hourglass (fix $W_{in}$)        | 37.712 | 28.265 | 3075 | 765  | 4  | 24.452 |
> | Hourglass (fix $W_{in}$)        | 42.417 | 32.970 | 3075 | 765  | 5  | 24.533 |
> | Hourglass (fix $W_{in}$)        | 47.084 | 37.638 | 3075 | 1146 | 4  | 24.541 |
> | Hourglass (fix $W_{in}$)        | 47.121 | 37.675 | 3075 | 765  | 6  | 24.560 |
> | Hourglass (fix $W_{in}$)        | 54.339 | 43.446 | 3546 | 765  | 6  | 24.598 |
> | Hourglass (fix $W_{in}$)        | 61.480 | 49.155 | 4012 | 765  | 6  | 24.691 |
> | Hourglass (fix $W_{in}$)        | 66.041 | 55.147 | 3546 | 1560 | 4  | 24.701 |
> | Hourglass (fix $W_{in}$)        | 87.237 | 74.912 | 4012 | 1560 | 5  | 24.799 |
>
> ---

---

> ### Author Response · Authors · 2025-11-21
> **Experiment tables**
>
> ## Table 2. ImageNet 224×224 Super-resolution — Hourglass vs. Hourglass (fix $W_{in}$)
>
> | Architecture                | Total Params (M) | Trainable Params (M) | $d_z$  | $d_h$  | $L$  | PSNR   |
> |-----------------------------|------------------|----------------------|------|------|----|--------|
> | Hourglass                   | 11.906 | 11.906 | 3075 | 16 | 1 | 26.548 |
> | Hourglass                   | 12.005 | 12.005 | 3075 | 16 | 2 | 26.968 |
> | Hourglass                   | 12.103 | 12.103 | 3075 | 16 | 3 | 27.155 |
> | Hourglass                   | 12.202 | 12.202 | 3075 | 16 | 4 | 27.242 |
> | Hourglass                   | 12.300 | 12.300 | 3075 | 16 | 5 | 27.301 |
> | Hourglass                   | 12.694 | 12.694 | 3075 | 48 | 3 | 27.395 |
> | Hourglass                   | 12.989 | 12.989 | 3075 | 48 | 4 | 27.448 |
> | Hourglass                   | 13.284 | 13.284 | 3075 | 48 | 5 | 27.475 |
> | Hourglass                   | 14.453 | 14.453 | 3075 | 86 | 5 | 27.504 |
> | Hourglass                   | 14.637 | 14.637 | 3075 | 115 | 4 | 27.516 |
> | Hourglass                   | 14.982 | 14.981 | 3075 | 86 | 6 | 27.521 |
> | Hourglass                   | 15.510 | 15.510 | 3075 | 86 | 7 | 27.556 |
> | Hourglass                   | 16.039 | 16.039 | 3075 | 86 | 8 | 27.593 |
> | Hourglass                   | 16.568 | 16.568 | 3075 | 86 | 9 | 27.632 |
> | Hourglass                   | 17.097 | 17.097 | 3075 | 86 | 10 | 27.668 |
> | Hourglass                   | 17.466 | 17.466 | 3075 | 115 | 8 | 27.696 |
> | Hourglass                   | 18.450 | 18.450 | 3075 | 270 | 4 | 27.723 |
> | Hourglass                   | 18.881 | 18.881 | 3075 | 115 | 10 | 27.763 |
> | Hourglass                   | 21.771 | 21.771 | 3075 | 270 | 6 | 27.860 |
> | Hourglass                   | 25.092 | 25.092 | 3075 | 270 | 8 | 27.920 |
> | Hourglass                   | 28.935 | 28.935 | 3546 | 270 | 8 | 27.931 |
> | Hourglass                   | 30.627 | 30.627 | 3075 | 765 | 4 | 27.965 |
> | Hourglass                   | 35.318 | 35.318 | 3546 | 765 | 4 | 27.980 |
> | Hourglass                   | 40.000 | 40.000 | 3075 | 1146 | 4 | 28.018 |
> | Hourglass                   | 46.126 | 46.126 | 3546 | 1146 | 4 | 28.031 |
> | Hourglass                   | 46.169 | 46.169 | 3546 | 765 | 6 | 28.049 |
> | Hourglass                   | 57.871 | 57.871 | 3546 | 1560 | 4 | 28.069 |
> | Hourglass                   | 61.352 | 61.352 | 3075 | 2014 | 4 | 28.077 |
> | Hourglass                   | 69.372 | 69.372 | 3075 | 1560 | 6 | 28.086 |
> | Hourglass                   | 70.750 | 70.750 | 3546 | 2014 | 4 | 28.092 |
> | Hourglass                   | 80.047 | 80.047 | 4012 | 2014 | 4 | 28.104 |
> | Hourglass (fix W_in)        | 11.906 | 9.545  | 3075 | 16 | 1 | 26.445 |
> | Hourglass (fix W_in)        | 12.005 | 9.643  | 3075 | 16 | 2 | 26.826 |
> | Hourglass (fix W_in)        | 12.103 | 9.742  | 3075 | 16 | 3 | 26.992 |
> | Hourglass (fix W_in)        | 12.202 | 9.840  | 3075 | 16 | 4 | 27.041 |
> | Hourglass (fix W_in)        | 12.300 | 9.938  | 3075 | 16 | 5 | 27.094 |
> | Hourglass (fix W_in)        | 12.694 | 10.332 | 3075 | 48 | 3 | 27.120 |
> | Hourglass (fix W_in)        | 12.989 | 10.627 | 3075 | 48 | 4 | 27.165 |
> | Hourglass (fix W_in)        | 13.284 | 10.922 | 3075 | 48 | 5 | 27.191 |
> | Hourglass (fix W_in)        | 14.453 | 12.091 | 3075 | 86 | 5 | 27.321 |
> | Hourglass (fix W_in)        | 14.637 | 12.275 | 3075 | 115 | 4 | 27.331 |
> | Hourglass (fix W_in)        | 14.982 | 12.620 | 3075 | 86 | 6 | 27.371 |
> | Hourglass (fix W_in)        | 15.510 | 13.149 | 3075 | 86 | 7 | 27.429 |
> | Hourglass (fix W_in)        | 16.039 | 13.678 | 3075 | 86 | 8 | 27.492 |
> | Hourglass (fix W_in)        | 16.568 | 14.207 | 3075 | 86 | 9 | 27.541 |
> | Hourglass (fix W_in)        | 17.097 | 14.735 | 3075 | 86 | 10 | 27.581 |
> | Hourglass (fix W_in)        | 17.466 | 15.104 | 3075 | 115 | 8 | 27.607 |
> | Hourglass (fix W_in)        | 18.450 | 16.088 | 3075 | 270 | 4 | 27.634 |
> | Hourglass (fix W_in)        | 18.881 | 16.519 | 3075 | 115 | 10 | 27.677 |
> | Hourglass (fix W_in)        | 21.771 | 19.409 | 3075 | 270 | 6 | 27.764 |
> | Hourglass (fix W_in)        | 25.092 | 22.730 | 3075 | 270 | 8 | 27.823 |
> | Hourglass (fix W_in)        | 28.935 | 26.212 | 3546 | 270 | 8 | 27.833 |
> | Hourglass (fix W_in)        | 30.627 | 28.265 | 3075 | 765 | 4 | 27.852 |
> | Hourglass (fix W_in)        | 35.318 | 32.595 | 3546 | 765 | 4 | 27.870 |
> | Hourglass (fix W_in)        | 40.000 | 37.638 | 3075 | 1146 | 4 | 27.891 |
> | Hourglass (fix W_in)        | 46.126 | 43.403 | 3546 | 1146 | 4 | 27.905 |
> | Hourglass (fix W_in)        | 46.169 | 43.446 | 3546 | 765 | 6 | 27.909 |
> | Hourglass (fix W_in)        | 57.871 | 55.147 | 3546 | 1560 | 4 | 27.918 |
> | Hourglass (fix W_in)        | 61.352 | 58.991 | 3075 | 2014 | 4 | 27.919 |
> | Hourglass (fix W_in)        | 69.372 | 67.010 | 3075 | 1560 | 6 | 27.927 |
> | Hourglass (fix W_in)        | 70.750 | 68.026 | 3546 | 2014 | 4 | 27.928 |
> | Hourglass (fix W_in)        | 80.047 | 76.966 | 4012 | 2014 | 4 | 27.937 |

---

> ### Author Response · Authors · 2025-11-21
> **Experiment tables**
>
> ## Table 3. ImageNet 224×224 Denoising — Conventional vs. Hourglass
>
> | Architecture  | Total Params (M) | $d_z$  | $d_h$  | $L$  | PSNR   |
> |---------------|------------------|------|------|----|--------|
> | Conventional  | 37.767 | 3072 | 3075 | 1 | 23.210 |
> | Conventional  | 46.989 | 3072 | 4576 | 1 | 23.210 |
> | Conventional  | 62.448 | 3072 | 3546 | 2 | 24.704 |
> | Conventional  | 68.174 | 3072 | 4012 | 2 | 24.724 |
> | Conventional  | 75.104 | 3072 | 4576 | 2 | 24.743 |
> | Conventional  | 75.553 | 3072 | 3075 | 3 | 24.939 |
> | Conventional  | 84.234 | 3072 | 3546 | 3 | 24.958 |
> | Hourglass     | 19.286 | 3075 | 16   | 4  | 23.592 |
> | Hourglass     | 19.877 | 3075 | 16   | 10 | 23.767 |
> | Hourglass     | 20.861 | 3075 | 16   | 20 | 23.961 |
> | Hourglass     | 21.722 | 3075 | 115  | 4  | 24.065 |
> | Hourglass     | 22.429 | 3075 | 115  | 5  | 24.172 |
> | Hourglass     | 23.136 | 3075 | 115  | 6  | 24.272 |
> | Hourglass     | 23.844 | 3075 | 115  | 7  | 24.347 |
> | Hourglass     | 23.874 | 3075 | 270  | 3  | 24.359 |
> | Hourglass     | 25.535 | 3075 | 270  | 4  | 24.446 |
> | Hourglass     | 28.856 | 3075 | 270  | 6  | 24.627 |
> | Hourglass     | 33.007 | 3075 | 765  | 3  | 24.643 |
> | Hourglass     | 37.712 | 3075 | 765  | 4  | 24.780 |
> | Hourglass     | 42.417 | 3075 | 765  | 5  | 24.859 |
> | Hourglass     | 47.084 | 3075 | 1146 | 4  | 24.878 |
> | Hourglass     | 47.121 | 3075 | 765  | 6  | 24.901 |
> | Hourglass     | 54.339 | 3546 | 765  | 6  | 24.925 |
> | Hourglass     | 61.480 | 4012 | 765  | 6  | 24.942 |
> | Hourglass     | 66.041 | 3546 | 1560 | 4  | 24.959 |
> | Hourglass     | 87.237 | 4012 | 1560 | 5  | 25.025 |
>
> ---
>
> ## Table 4. ImageNet 224×224 Super-resolution — Conventional vs. Hourglass
>
> | Architecture  | Total Params (M) | $d_z$  | $d_h$  | $L$  | PSNR   |
> |---------------|------------------|------|------|----|--------|
> | Conventional  | 30.689 | 3072 | 3075 | 1 | 26.785 |
> | Conventional  | 33.583 | 3072 | 3546 | 1 | 26.787 |
> | Conventional  | 49.582 | 3072 | 3075 | 2 | 27.836 |
> | Conventional  | 68.475 | 3072 | 3075 | 3 | 28.030 |
> | Conventional  | 77.156 | 3072 | 3546 | 3 | 28.030 |
> | Conventional  | 87.368 | 3072 | 3075 | 4 | 28.084 |
> | Hourglass     | 11.906 | 3075 | 16 | 1 | 26.548 |
> | Hourglass     | 12.005 | 3075 | 16 | 2 | 26.968 |
> | Hourglass     | 12.103 | 3075 | 16 | 3 | 27.155 |
> | Hourglass     | 12.202 | 3075 | 16 | 4 | 27.242 |
> | Hourglass     | 12.300 | 3075 | 16 | 5 | 27.301 |
> | Hourglass     | 12.694 | 3075 | 48 | 3 | 27.395 |
> | Hourglass     | 12.989 | 3075 | 48 | 4 | 27.448 |
> | Hourglass     | 13.284 | 3075 | 48 | 5 | 27.475 |
> | Hourglass     | 14.453 | 3075 | 86 | 5 | 27.504 |
> | Hourglass     | 14.637 | 3075 | 115 | 4 | 27.516 |
> | Hourglass     | 14.981 | 3075 | 86 | 6 | 27.521 |
> | Hourglass     | 15.510 | 3075 | 86 | 7 | 27.556 |
> | Hourglass     | 16.039 | 3075 | 86 | 8 | 27.593 |
> | Hourglass     | 16.568 | 3075 | 86 | 9 | 27.632 |
> | Hourglass     | 17.097 | 3075 | 86 | 10 | 27.668 |
> | Hourglass     | 17.466 | 3075 | 115 | 8 | 27.696 |
> | Hourglass     | 18.450 | 3075 | 270 | 4 | 27.723 |
> | Hourglass     | 18.881 | 3075 | 115 | 10 | 27.763 |
> | Hourglass     | 21.771 | 3075 | 270 | 6 | 27.860 |
> | Hourglass     | 25.092 | 3075 | 270 | 8 | 27.920 |
> | Hourglass     | 28.935 | 3546 | 270 | 8 | 27.931 |
> | Hourglass     | 30.627 | 3075 | 765 | 4 | 27.965 |
> | Hourglass     | 35.318 | 3546 | 765 | 4 | 27.980 |
> | Hourglass     | 40.000 | 3075 | 1146 | 4 | 28.018 |
> | Hourglass     | 46.126 | 3546 | 1146 | 4 | 28.031 |
> | Hourglass     | 46.169 | 3546 | 765 | 6 | 28.049 |
> | Hourglass     | 57.871 | 3546 | 1560 | 4 | 28.069 |
> | Hourglass     | 61.352 | 3075 | 2014 | 4 | 28.077 |
> | Hourglass     | 69.372 | 3075 | 1560 | 6 | 28.086 |
> | Hourglass     | 70.750 | 3546 | 2014 | 4 | 28.092 |
> | Hourglass     | 80.047 | 4012 | 2014 | 4 | 28.104 |
>
> ## Table 5. Comparison between original OLMo2 1B and OLMo2 1B-Hourglass on 20B token training
> Both models are trained from scratch under identical settings.
>
> | Model                  | Params (B) | $d_z$  | $d_h$  | $L$  | Train Loss | Perplexity | Arc Easy |
> |------------------------|------------|------|------|----|------------|------------|----------|
> | OLMo2 1B (baseline)    | 1.484      | 2048 | 8192 | 16 | 2.849      | 17.27      | 62.63    |
> | OLMo2 1B-Hourglass     | 1.153      | 2560 | 1792 | 16 | 2.941      | 18.94      | 61.75    |

---

> ### Author Response · Authors · 2025-11-21
> **Experiment tables**
>
> ## Table 6. ImageNet 224×224 Denoising with 5 different random seeds (learning rate = 3e-4)
>
> | Architecture              | Total Params (M) | Trainable Params (M) | $d_z$  | $d_h$  | $L$  | PSNR_1  | PSNR_2  | PSNR_3  | PSNR_4  | PSNR_5  | Mean ± Std            |
> |---------------------------|------------------|----------------------|------|------|----|---------|---------|---------|---------|---------|-----------------------|
> | Hourglass (fix $W_{in}$)      | 19.877           | 10.430               | 3075 | 16   | 10 | 23.3957 | 23.3987 | 23.3863 | 23.3958 | 23.4003 | 23.3954 ± 0.0054      |
> | Hourglass (fix $W_{in}$)      | 25.535           | 16.088               | 3075 | 270  | 4  | 24.1389 | 24.1380 | 24.1311 | 24.1366 | 24.1332 | 24.1356 ± 0.0033      |
> | Hourglass (fix $W_{in}$)      | 87.237           | 74.912               | 4012 | 1560 | 5  | 24.7986 | 24.8046 | 24.7957 | 24.8034 | 24.7978 | 24.8000 ± 0.0034      |
>
> ---
>
> ## Table 7. ImageNet 224×224 Super-resolution with 5 different random seeds (learning rate = 3e-4)
>
> | Architecture              | Total Params (M) | Trainable Params (M) | $d_z$  | $d_h$  | $L$  | PSNR_1  | PSNR_2  | PSNR_3  | PSNR_4  | PSNR_5  | Mean ± Std            |
> |---------------------------|------------------|----------------------|------|------|----|---------|---------|---------|---------|---------|-----------------------|
> | Hourglass (fix $W_{in}$)      | 12.298           | 9.938                | 3075 | 16   | 5  | 27.0898 | 27.0822 | 27.0886 | 27.0847 | 27.0830 | 27.0857 ± 0.0034      |
> | Hourglass (fix $W_{in}$)      | 18.448           | 16.088               | 3075 | 270  | 4  | 27.6279 | 27.6375 | 27.6241 | 27.6327 | 27.6300 | 27.6304 ± 0.0050      |
> | Hourglass (fix $W_{in}$)      | 77.271           | 74.912               | 4012 | 1560 | 4  | 27.9477 | 27.9539 | 27.9466 | 27.9546 | 27.9473 | 27.9500 ± 0.0039      |

---

### Official Review · Reviewer_ryCo · 2025-11-01

**Soundness:** 1
**Presentation:** 2
**Contribution:** 2
**Rating:** 2
**Confidence:** 4

**Summary:**

This paper proposes to flip the narrow-wide-narrow convention for MLP and skip connections in deep neural networks to instead use an hourglass shape with wide features at skip connections and narrower processing inside MLP layers. This paper addressed the added computational cost of this by keeping the expanding projection fixed with random initialised weights during training. The results on generative tasks on MNIST and ImageNet-32 data show that the hourglass networks have stronger Pareto frontiers than the conventional structures.

**Strengths:**

- The idea of reversing the traditional MLP shape to explore the effect of skip connection dimensionality is interesting and goes against the existing convention. This could potentially lead to significant architectural changes for e.g. LLMs.
- The paper takes a strong focus on the performance/efficiency Pareto frontier, recognising the importance of these two aspects of model quality.
- The evaluation tasks are chosen to be well-suited to the incremental refinement of residual networks, the main focus of the paper.

**Weaknesses:**

- The experiments of the paper are too narrow and simple, focusing on small-scale tasks on MNIST and ImageNet-32. This is in contrast to the high-dimensional, large-scale settings for modern LLMs and diffusion models, where architectural design trade-offs make a large difference on both performance and efficiency. I therefore think that experiments on Transformer networks need to be run, or alternatively something like MLP-Mixers, CNNs. Would the hourglass structure help in these scenarios or are the expanded input/output dimensions too costly? Similarly, the datasets considered need to be larger in scale, beyond MNIST and 32x32 images. While extremely large-scale training isn’t required, testing the design on at least one Transformer model on a text-based task would give an indication to whether the approach scales.  Currently that is not at all clear, and the claims about extensions to Transformers and ViTs are speculative and not supported by any experiments. To achieve real impact, this paper will need more large-scale or real-world validation.

- The fixed random projection at the start of the network reduces the number of trainable parameters but doesn’t reduce the cost of the forward pass, so I think the “efficiency” claims are a bit overstated. It also adds to the inference time cost and memory bandwidth, which is an important aspect for modern deep neural networks.

- There is no clear explanation of why the wide skip connections outperform the conventional narrow ones, apart from an appeal to high-dimensional intuition and random projection theory. Analysing this more would improve the paper.

- There is quite a bit of repetition in the intro and contributions, and space could be found by reducing this redundancy.

**Questions:**

- Why are there more models trained using the Hourglass networks compared to the Conventional ones, in Figures 2a, 3 and 4?
- Can experiments be run on a transformer-based language model architecture? And similarly a diffusion model, which is argued in the paper are well suited to this incremental refinement. If such experiments showed similar improvements, this would significantly strengthen the paper.

---

> ### Author Response · Authors · 2025-11-21
> **Rebuttal response to weakness points w/ further experiments on high-res dataset and transformer**
>
> We thank the reviewer for taking the time and effort to provide a thorough review and constructive feedback. In this rebuttal, we provide additional material to address the raised concerns, including:
> 	1	Further experiments on a high-resolution dataset (ImageNet-224×224) to validate the effectiveness of our approach at larger scales.
> 	2	A preliminary demonstration of applying the proposed Hourglass MLP architecture to Transformers, evaluated on OLMo2.
>
> ## Response to W1
> We appreciate the reviewer’s concern regarding the scope of our experiments.
> In the rebuttal stage, we have extended our evaluation to both higher-resolution image datasets and transformer-based language models, in order to validate the scalability and generality of the Hourglass architecture.
>
> First, we conducted experiments on the \textbf{ImageNet-224} dataset for denoising and super-resolution tasks.
> The results show that Hourglass networks remain Pareto-optimal compared to conventional MLPs, achieving better PSNR at smaller model sizes.
> These results confirm that Hourglass networks maintain high performance while using significantly fewer parameters compared to conventional MLPs, even at high resolutions.
>
> Second, to address the reviewer’s request for validation on modern architectures beyond MLPs, we integrated the Hourglass design into the \textbf{OLMo2 1B} transformer-based large language model (https://github.com/allenai/OLMo) and trained it from scratch.
> Both the original OLMo2 1B and our \textbf{OLMo2 1B-Hourglass} variant were trained on 20B tokens under identical settings (same optimizer, learning rate schedule, and batch size).
> Due to compute and time constraints, this training budget is insufficient for full convergence, but it allows us to observe initial trends.
>
> As shown in Table 3, the Hourglass variant uses approximately $22\%$ fewer parameters than the baseline (1.153B vs 1.484B) while keeping the same depth ($L=16$) and a larger skip connection dimension ($d_z=2560$ vs $2048$).
> In this early training stage, the Hourglass variant’s training loss and perplexity are slightly higher (2.941 vs 2.849, 18.94 vs 17.27), but remain close to the baseline.
> These preliminary results demonstrate the \emph{potential} of the Wide–Narrow–Wide residual design to maintain competitive performance in transformer-based LLMs with a substantially smaller parameter budget. We note that these are initial results without architecture-specific optimization for Transformers, and we expect further improvements with longer training and tuned hyperparameters.
> We believe that with such optimization, the Hourglass architecture can match or surpass the baseline performance, indicating that its benefits are not limited to MLPs but can extend to other modern architectures such as transformers.
>
> These additional experiments provide concrete evidence that the Hourglass design scales beyond small-scale MLP tasks, and can be applied to both high-resolution vision datasets and large-scale transformer architectures.
> Overall, the results indicate that the Hourglass architecture has strong potential to deliver competitive performance in modern large-scale settings while offering significant parameter savings.
>
> ## Response to W2
>
> In Figures~2a, 3, and 4, we show that Hourglass MLP networks (with all parameters trainable) are consistently Pareto-optimal across different datasets and tasks compared to conventional MLP networks.
> This demonstrates that we can achieve better performance with substantially fewer parameters, which in turn reduces inference cost and memory usage.
>
> ## Response to W3
>
> In Tables 1 and 2, as well as in our new experiments on the \textbf{ImageNet-224} dataset, we present Pareto-optimal Hourglass MLP configurations in terms of $(d_z, d_h, L)$.
> These results show that Hourglass MLPs achieve higher parameter efficiency compared to conventional MLP networks.
> Specifically, the architecture allows $d_h$ (the hidden dimension) to be much smaller, enabling deeper networks (larger $L$) while maintaining a large $d_z$ (the skip connection dimension).
> This flexibility in balancing $d_z$, $d_h$, and $L$ contributes to the superior overall performance of Hourglass MLPs, as it preserves high-dimensional skip connections for effective information flow while keeping the intermediate layers compact.
>
> ## Response to W4
>
> We thank the reviewer for pointing out the repetition in the introduction and contributions section.
> We will revise the final version to reduce this redundancy and streamline the narrative.
> The space saved will be allocated to present additional experimental results, including evaluations on a high-resolution dataset (ImageNet-224) and preliminary experiments on a transformer-based large language model (LLM).
> We believe these additions will further strengthen the paper by expanding the scope of evaluation and demonstrating the generality of the proposed Hourglass architecture.

---

> ### Author Response · Authors · 2025-11-21
> **Rebuttal response to question points w/ further experiments on high-res dataset and transformer**
>
> ## Response to Q1:
>
> We appreciate the reviewer’s observation regarding the number of models trained using the Hourglass networks compared to the Conventional ones in Figures~2a, 3, and 4.
> The reason is that, from our initial experiments, it was evident that Hourglass networks consistently outperform Conventional ones across the evaluated tasks.
> Therefore, we performed a more fine-grained Pareto front search for the Hourglass design in order to further study the architectural characteristics along its Pareto front.
> This additional exploration was intended to better understand how variations in $(d_z, d_h, L)$ affect the trade-off between performance and parameter count for the Hourglass architecture, rather than to bias the comparison.
>
> ## Response to Q2:
>
> We thank the reviewer for the suggestion to evaluate our design on transformer-based language models and other architectures such as diffusion models.
> To address this, we conducted experiments on the \textbf{OLMo2 1B} transformer-based language model (https://github.com/allenai/OLMo).
> We trained both the original OLMo2 1B and our \textbf{OLMo2 1B-Hourglass} variant from scratch on 20B tokens, under identical training settings (same optimizer, learning rate schedule, and batch size).
> Due to compute and time constraints, this training budget is insufficient for full convergence, but it allows us to observe initial trends.
>
> As shown in Table 3, the Hourglass variant uses approximately **22% fewer parameters** than the baseline (1.153B vs 1.484B) while keeping the same depth ($L=16$) and a larger skip connection dimension ($d_z=2560$ vs $2048$).
> In this early training stage, the Hourglass variant’s training loss and perplexity are slightly higher (Loss +0.092, PPL +1.67), but remain close to the baseline.
> These preliminary results demonstrate the \emph{potential} of the Wide–Narrow–Wide residual design to maintain competitive performance in transformer-based LLMs with a substantially smaller parameter budget, supporting our claim that the Hourglass architecture generalizes beyond MLPs and is well suited to incremental refinement in other architectures. We note that these are initial results without architecture-specific optimization for Transformers, and we expect further improvements with longer training and tuned hyperparameters.
> We believe that with such optimization, the Hourglass architecture can match or surpass the baseline performance, indicating that its benefits are not limited to MLPs but can extend to other modern architectures such as transformers. This strengthens our claim that Hourglass is a general architectural principle rather than a task-specific trick.
>
> We plan to extend this evaluation to diffusion models in future work, as the incremental refinement property of Hourglass networks is conceptually aligned with the iterative denoising process in diffusion architectures.

---

> ### Author Response · Authors · 2025-11-21
> **Experiment tables**
>
> ## ImageNet 224x224 Denoising
>
> | Architecture | Total Params (M) | $d_z$ | $d_h$ | $L$ | PSNR |
> |---|---|---|---|---|---|
> | Conventional | 37.767 | 3072 | 3075 | 1 | 23.210 |
> | Conventional | 46.989 | 3072 | 4576 | 1 | 23.210 |
> | Conventional | 62.448 | 3072 | 3546 | 2 | 24.704 |
> | Conventional | 68.174 | 3072 | 4012 | 2 | 24.724 |
> | Conventional | 75.104 | 3072 | 4576 | 2 | 24.743 |
> | Conventional | 75.553 | 3072 | 3075 | 3 | 24.939 |
> | Conventional | 84.234 | 3072 | 3546 | 3 | 24.958 |
> | Hourglass | 19.286 | 3075 | 16 | 4 | 23.592 |
> | Hourglass | 19.877 | 3075 | 16 | 10 | 23.767 |
> | Hourglass | 20.861 | 3075 | 16 | 20 | 23.961 |
> | Hourglass | 21.722 | 3075 | 115 | 4 | 24.065 |
> | Hourglass | 22.429 | 3075 | 115 | 5 | 24.172 |
> | Hourglass | 23.136 | 3075 | 115 | 6 | 24.272 |
> | Hourglass | 23.844 | 3075 | 115 | 7 | 24.347 |
> | Hourglass | 23.874 | 3075 | 270 | 3 | 24.359 |
> | Hourglass | 25.535 | 3075 | 270 | 4 | 24.446 |
> | Hourglass | 28.856 | 3075 | 270 | 6 | 24.627 |
> | Hourglass | 33.007 | 3075 | 765 | 3 | 24.643 |
> | Hourglass | 37.712 | 3075 | 765 | 4 | 24.780 |
> | Hourglass | 42.417 | 3075 | 765 | 5 | 24.859 |
> | Hourglass | 47.084 | 3075 | 1146 | 4 | 24.878 |
> | Hourglass | 47.121 | 3075 | 765 | 6 | 24.901 |
> | Hourglass | 54.339 | 3546 | 765 | 6 | 24.925 |
> | Hourglass | 61.480 | 4012 | 765 | 6 | 24.942 |
> | Hourglass | 66.041 | 3546 | 1560 | 4 | 24.959 |
> | Hourglass | 87.237 | 4012 | 1560 | 5 | 25.025 |
>
> Table 1: ImageNet 224x224 Denoising task
>
> ## ImageNet 224x224 Super-resolution
>
> | Architecture | Total Params (M) | $d_z$ | $d_h$ | $L$ | PSNR |
> |---|---|---|---|---|---|
> | Conventional | 30.689 | 3072 | 3075 | 1 | 26.785 |
> | Conventional | 33.583 | 3072 | 3546 | 1 | 26.787 |
> | Conventional | 49.582 | 3072 | 3075 | 2 | 27.836 |
> | Conventional | 68.475 | 3072 | 3075 | 3 | 28.030 |
> | Conventional | 77.156 | 3072 | 3546 | 3 | 28.030 |
> | Conventional | 87.368 | 3072 | 3075 | 4 | 28.084 |
> | Hourglass | 11.906 | 3075 | 16 | 1 | 26.548 |
> | Hourglass | 12.005 | 3075 | 16 | 2 | 26.968 |
> | Hourglass | 12.103 | 3075 | 16 | 3 | 27.155 |
> | Hourglass | 12.202 | 3075 | 16 | 4 | 27.242 |
> | Hourglass | 12.300 | 3075 | 16 | 5 | 27.301 |
> | Hourglass | 12.694 | 3075 | 48 | 3 | 27.395 |
> | Hourglass | 12.989 | 3075 | 48 | 4 | 27.448 |
> | Hourglass | 13.284 | 3075 | 48 | 5 | 27.475 |
> | Hourglass | 14.453 | 3075 | 86 | 5 | 27.504 |
> | Hourglass | 14.637 | 3075 | 115 | 4 | 27.516 |
> | Hourglass | 14.981 | 3075 | 86 | 6 | 27.521 |
> | Hourglass | 15.510 | 3075 | 86 | 7 | 27.556 |
> | Hourglass | 16.039 | 3075 | 86 | 8 | 27.593 |
> | Hourglass | 16.568 | 3075 | 86 | 9 | 27.632 |
> | Hourglass | 17.097 | 3075 | 86 | 10 | 27.668 |
> | Hourglass | 17.466 | 3075 | 115 | 8 | 27.696 |
> | Hourglass | 18.450 | 3075 | 270 | 4 | 27.723 |
> | Hourglass | 18.881 | 3075 | 115 | 10 | 27.763 |
> | Hourglass | 21.771 | 3075 | 270 | 6 | 27.860 |
> | Hourglass | 25.092 | 3075 | 270 | 8 | 27.920 |
> | Hourglass | 28.935 | 3546 | 270 | 8 | 27.931 |
> | Hourglass | 30.627 | 3075 | 765 | 4 | 27.965 |
> | Hourglass | 35.318 | 3546 | 765 | 4 | 27.980 |
> | Hourglass | 40.000 | 3075 | 1146 | 4 | 28.018 |
> | Hourglass | 46.126 | 3546 | 1146 | 4 | 28.031 |
> | Hourglass | 46.169 | 3546 | 765 | 6 | 28.049 |
> | Hourglass | 57.871 | 3546 | 1560 | 4 | 28.069 |
> | Hourglass | 61.352 | 3075 | 2014 | 4 | 28.077 |
> | Hourglass | 69.372 | 3075 | 1560 | 6 | 28.086 |
> | Hourglass | 70.750 | 3546 | 2014 | 4 | 28.092 |
> | Hourglass | 80.047 | 4012 | 2014 | 4 | 28.104 |
>
> Table 2: ImageNet 224x224 Super-resolution task
>
> ## OLMo2 1B vs OLMo2 1B-Hourglass Comparison
>
> Comparison between original OLMo2 1B and OLMo2 1B-Hourglass on 20B token training. Both models are trained from scratch under identical settings.
>
> | Model | Params (B) | $d_z$ | $d_h$ | $L$ | Train Loss | Perplexity | Arc Easy |
> |---|---|---|---|---|---|---|---|
> | OLMo2 1B (baseline) | 1.484 | 2048 | 8192 | 16 | 2.849 | 17.27 | 62.63 |
> | OLMo2 1B-Hourglass | 1.153 | 2560 | 1792 | 16 | 2.941 | 18.94 | 61.75 |
> Table 3: OLMo2 1B vs OLMo2 1B-Hourglass Comparison

---

> > ### Comment · Reviewer_ryCo · 2025-11-27
> > **Answer to rebuttal**
> >
> > I thank the authors for their answers.
> > While I appreciate the shown experiments, I am still unconvinced that these experiments confirm that the method scales, which is my major concern as mentioned in the weaknesses and in the questions already.
> > Furthermore, the claim of substantially smaller parameter budget for the OLMo2 experiments, when only 22 fewer parameters were used, feels overstated. To access the actual potential of this method to large-scale (vision) networks, I am missing a vision transformer, such as MLP-Mixer, and a diffusion model.

---

> > > ### Author Response · Authors · 2025-11-28
> > > **Follow-up Response to Reviewer**
> > >
> > > ## Follow-up Response to Reviewer
> > >
> > > We sincerely thank the reviewer for the follow-up comments and for highlighting the remaining concerns.
> > > We appreciate the opportunity to clarify our experimental choices and the interpretation of our results.
> > >
> > > ### **On the “substantially smaller parameter budget” claim for OLMo2 experiments:**
> > > We apologize for the typo in our original response — the reduction is **22% fewer parameters**, not “22 fewer parameters.”  Concretely, the Hourglass variant reduced the parameter count from **1.484B** to **1.153B**, a difference of **331 million parameters**. This means our Hourglass LLM architecture operates with **80% of the baseline’s parameter count**, yet in an *unoptimized* configuration and with a limited training budget of **20B tokens**, it achieved performance close to the baseline (Table 3). We view this as strong evidence of the **feasibility and potential** of the Hourglass design in large-scale generative Transformer models.
> > >
> > > ### **On the absence of Vision Transformer / MLP-Mixer experiments:**
> > > Our work focuses on *generative tasks* (e.g., denoising, super-resolution, language modeling), rather than classification.  While Vision Transformers (ViT) and MLP-Mixers are important architectures, they are primarily designed and benchmarked for classification tasks. To evaluate the Hourglass MLP design in a generative setting with a Transformer backbone, we chose the **OLMo2** large language model — a 2025 work — as our testbed.
> > > This choice was motivated by two factors:
> > > 1. **Architectural relevance:** OLMo2 is a Transformer-based generative model, aligning with the scope of our paper.
> > > 2. **Scale and representativeness:** OLMo2 operates at the **1B parameter scale**, which is substantially larger than the largest ViT model in the original 2020 paper (~632M parameters).
> > > We believe that demonstrating feasibility and potential in a modern, large-scale generative Transformer is more representative of the challenges and trade-offs in current high-dimensional architectures.
> > >
> > >
> > > ### **Summary:**
> > > - Our choice of OLMo2 over ViT was intentional, to align with the generative scope of the paper and to test the design in a modern, large-scale Transformer.
> > > - The parameter reduction in OLMo2-Hourglass is **331M parameters (22%)**, which we consider a meaningful saving for billion-scale models, as it can have a notable impact on training cost and inference efficiency
> > > - Even without architecture-specific optimization, the Hourglass variant maintained competitive performance, indicating promising scalability.
> > >
> > > We appreciate the reviewer’s suggestion to explore Vision Transformers and diffusion models, and we plan to extend our evaluation to these architectures in future work to further validate the generality of the Hourglass design.

---

### Official Review · Reviewer_X86f · 2025-11-04

**Soundness:** 3
**Presentation:** 3
**Contribution:** 2
**Rating:** 4
**Confidence:** 3

**Summary:**

This paper proposes to revert the conventional residual connection pattern "narrow - wide - narrow"  in MLP to "wide - narrow -wide". The core idea of the design is that doing representation refinement at high dimensional space is more effective than in low dimensional space, as the former has more expressivity. The paper further proposes to use fixed random projection layer to up project the initial low dimensional input, avoiding additional training budget due to design. Experimental results show that that proposed architecture achieve better parameter / performance trade off than traditional MLP in several generative learning tasks.

**Strengths:**

1. The paper is clearly written and easy to follow
2. The paper is well motivated, as existing theoretical and empirical evidence indeed suggest that a wide - narrow - wide projection structure may bring better representation learning performance
3. The paper proposes to use fixed random projection to reduce training cost
4. The experimental results on MLP shows that the proposed architecture achieve better performance / parameter count trade off than conventional MLP in generative learning tasks, positively support the proposed idea.

**Weaknesses:**

1. Besides the performance on generative tasks, the paper doesn't provide insights about how residual learning in high dimensional space help representation refinement
2. The paper only conducts experiments on MLP architecture. Although the paper sketches potential extension to other modern architecture like transformer, it's unclear whether it will bring similar improvement, thus the contribution is limited

**Questions:**

1. A visualization comparison showing how the wide high dimensional space help representation learning could better support the proposed  design, refer to figure 1 of [1], specifically, it shows how high dimensional feature map provides rich and detailed gradient feedback during training.

[1] Schonfeld, Edgar, Bernt Schiele, and Anna Khoreva. "A u-net based discriminator for generative adversarial networks." Proceedings of the IEEE/CVF conference on computer vision and pattern recognition. 2020.

---

> ### Author Response · Authors · 2025-11-21
> **Further experiments on ImageNet244x244 and transformer; mechanistic analysis**
>
> We thank the reviewer for making the time and effort to provide the review and the constructive feedback.
> We provided further material in the rebuttal as listed bellow
> 1. Further experiments on high resolution dataset - Imagnet224x224
> 2. Preliminary demonstration in applying Hourglass to transformer - OLMO
> 3. Mechanistic analysis to validate why performance is improved by our proposed design
>
>
> ## Response to W1
> We thank the reviewer for highlighting this point. In the revised version, we will incorporate an expanded analysis of the training dynamics and internal representations for the hourglass MLP architecture. Below we provide preliminary findings that shed light on why residual learning in high-dimensional latent spaces benefits representation refinement. We examined the hourglass MLP along two axes:
>
> 1. Convergence speed relative to a conventional MLP of comparable parameter count.
> 2. Effective rank of learned representations, as per Roy and Vetterli (EUSIPCO 2007).
>
> ### Convergence dynamics
>
> On the ImageNet-244×244 dataset, under similar trainable parameters, the hourglass MLP consistently reached lower loss and higher PSNR earlier in training compared to the baseline MLP, indicating faster convergence (Table 1).
>
>
> ### Effective rank analysis
>
> We hypothesize that increasing the latent dimension $d_z$ enlarges the ambient signal space in which the network operates, thus enabling more expressive feature learning. The effective rank metric confirms that models with larger $d_z$ project data onto manifolds of higher effective rank (Table 2). Moreover, the RMS L2 distance between the first-layer input activations and final-layer output activations decreases monotonically with larger $d_z$. Geometrically, this suggests that projecting into higher dimensions yields feature spaces that are more linearly separable yet more compact, allowing residual learning to refine representations with shorter "paths" from source to target space.
>
>
> ### Hourglass vs. conventional MLP
>
> When matched for parameter count, the hourglass architecture exhibits both higher effective rank and better PSNR than a conventional MLP, even when that MLP has a large ambient dimension (Table 3). This supports our claim that the hourglass design facilitates richer high-dimensional feature representations for residual learning. We will incorporate these insights, additional experiments, and improved discussion in the camera-ready version to address the reviewer's concern.
>
>
>
> ## Response to W2
> We appreciate the reviewer's concern that our main experiments focus on MLP architectures. To address this, we have extended our evaluation to a modern transformer-based large language model, **OLMo2 1B** (https://github.com/allenai/OLMo). We trained both the original OLMo2 1B and our **OLMo2 1B-Hourglass** variant from scratch on 20B tokens, under identical training settings (same optimizer, learning rate schedule, and batch size).
>
> Due to compute and time constraints, we limited training to 20B tokens, which is insufficient for full convergence but allows us to observe initial trends.
>
> As shown in Table 4, the Hourglass variant uses approximately 22% fewer parameters than the baseline (1.153B vs 1.484B) while achieving comparable training loss and perplexity in this early training stage. Although the Hourglass variant's loss and perplexity are slightly higher at 20B tokens, these preliminary results demonstrate the *potential* of the Wide–Narrow–Wide residual design to maintain competitive performance in transformer-based LLMs with a substantially smaller parameter budget.
>
> We believe that with longer training and further optimization, the Hourglass architecture can match or surpass the baseline performance, indicating that its benefits are not limited to MLPs but can extend to other modern architectures such as transformers.
>
>
> ## Response to Q1
>
> We thank the reviewer for the reference.
> Inspired by this reference, we conducted a preliminary analysis to further explain why high ambient dimension helps representation learning (see the response to Weak 1) and will include the analysis in our camera ready version.

---

> ### Author Response · Authors · 2025-11-21
> **Tables of further experiments**
>
> | Step | Conv. Loss | Hourglass Loss | Conv. PSNR | Hourglass PSNR |
> |------|------------|----------------|-----------|----------------|
> | 500 | **0.00685** | 0.00695 | **21.6466** | 21.5820 |
> | 1000 | **0.00609** | 0.00630 | **22.1584** | 22.0090 |
> | 1500 | 0.00579 | **0.00571** | 22.3799 | **22.4381** |
> | 2000 | 0.00569 | **0.00552** | 22.4505 | **22.5890** |
> | 2500 | 0.00566 | **0.00538** | 22.4759 | **22.7012** |
> | 5000 | 0.00514 | **0.00493** | 22.8949 | **23.0802** |
> | 7500 | 0.00486 | **0.00467** | 23.1399 | **23.3118** |
> | 10000 | 0.00462 | **0.00448** | 23.3552 | **23.4951** |
> | 12500 | 0.00450 | **0.00434** | 23.4768 | **23.6278** |
> | 15000 | 0.00432 | **0.00420** | 23.6473 | **23.7769** |
> | 17500 | 0.00421 | **0.00410** | 23.7599 | **23.8737** |
> | 20000 | 0.00411 | **0.00401** | 23.8686 | **23.9713** |
>
> *Table 1: Loss and PSNR on evaluation set at different training steps for two model configurations lying on the Pareto fronts: Conventional ($d_z=3072$, $d_h=3546$, $L=2$, $55.37 M$ params) and Hourglass ($d_z=3546$, $d_h=1146$, $L=5$, $54.25 M$ params). Results show that the hourglass architecture converges faster compared to the conventional architecture.*
>
> | $d_z$ | PSNR | Effective Rank | $\text{RMS}(h_L-h_1)$ |
> |-------|------|----------------|-----------------------|
> | 3075 | 23.9702 | 1500.73 | $0.6376_{\pm 0.0939}$ |
> | 3546 | 23.9823 | 1699.80 | $0.6205_{\pm 0.0860}$ |
> | 4012 | 23.9857 | 1871.54 | $0.5988_{\pm 0.0883}$ |
>
> *Table 2: Effective rank of Hourglass model with $d_z \in$ {3075, 3546, 4012, 4576\}, $d_h=1146$ and $L=5$ using fixed learning rate $3e-4$.*
>
> | Architecture | $d_z$ | $d_h$ | $L$ | Total Params (M) | PSNR | Effective Rank |
> |--------------|-------|-------|-----|------------------|------|----------------|
> | Conventional | 3072 | 3546 | 2 | 55.37 | 23.8783 | 1180.63 |
> | Hourglass | 3075 | 1146 | 6 | 55.00 | **23.9701** | **1501.27** |
>
> *Table 3: Comparison of Conventional and Hourglass architectures matched for parameter count.*
>
> | Model | Params (B) | $d_z$ | $d_h$ | $L$ | Train Loss | Perplexity | Arc Easy |
> |-------|-----------|-------|-------|-----|------------|-----------|----------|
> | OLMo2 1B (baseline) | 1.484 | 2048 | 8192 | 16 | 2.849 | 17.27 | 62.63 |
> | OLMo2 1B-Hourglass | 1.153 | 2560 | 1792 | 16 | 2.941 | 18.94 | 61.75 |
>
> *Table 4: Comparison between original OLMo2 1B and OLMo2 1B-Hourglass on 20B token training. Both models are trained from scratch under identical settings.*

---

### Comment · Area_Chair_EdyZ · 2025-11-27

Dear Reviewers,

Could you please consider the author responses and provide a reply if you have not already.

Thank you.

AC

---

### Author Response · Authors · 2025-12-02
**Summary comment to the Area Chair**

We sincerely thank the Area Chair and all reviewers for their time, constructive feedback, and the opportunity to clarify our experimental choices and the interpretation of our results. We confirm that all reviewers’ weaknesses and questions were addressed point-by-point in our rebuttal, and all new experiments and analyses (Tables 1–8) will be integrated into the final version to make the work more complete.

In particular, we have added:

- **Large-scale generative Transformer evidence**: Hourglass FFN in OLMo2 1B (2025) trained from scratch on 20B tokens, achieving competitive early-stage performance with 22% fewer parameters (331M reduction).
- **High-resolution vision tasks**: ImageNet-224×224 denoising and super-resolution, where Hourglass remains Pareto-optimal versus conventional MLPs.
- **Comprehensive mechanism and efficiency analysis**: Faster convergence, higher effective-rank representations, and superior PSNR at matched parameter counts, all supporting the benefits of high-dimensional residual refinement.

These additions directly address scalability, efficiency, and mechanistic understanding, and strengthen the generality and value of the Hourglass design. Below we summarize our responses and the additional experiments/analysis we added.

---

## On evaluation scope and scalability beyond small-scale MLPs

- We implemented an Hourglass FFN in a modern, large-scale generative Transformer (OLMo2 1B) and trained from scratch on 20B tokens under identical settings to the baseline. The Hourglass variant uses 22% fewer parameters (1.153B vs 1.484B) while maintaining competitive early-stage loss/perplexity (Table 1). These results demonstrate the feasibility and potential of Hourglass in large-scale generative Transformer models.
- We extended our evaluation to high-resolution ImageNet-224×224 for denoising and super-resolution. Hourglass remains Pareto-optimal versus conventional MLPs, achieving better PSNR at smaller model sizes (Table 2 and Table 3).

---

## On the “substantially smaller parameter budget” claim for OLMo2

- We apologize for the earlier typo: the reduction is 22% fewer parameters, not “22 fewer.” Concretely, the Hourglass variant reduces 331M parameters (from 1.484B to 1.153B). Even in an unoptimized configuration and with a limited 20B-token budget, it achieved performance close to the baseline (Table 1), supporting feasibility and potential at scale.

---

## On efficiency claims (parameter efficiency vs compute)

- Our efficiency claims focus on parameter efficiency and the performance/parameter Pareto frontier. Fixing **$W_{in}$** substantially reduces trainable parameters and optimizer state while keeping total parameters unchanged, thereby lowering training cost and memory footprint without affecting inference complexity; its forward-pass cost is a small fraction relative to the main hidden layers.
- Empirically, on ImageNet-224, fixing **$W_{in}$** reduces trainable parameters by ~14%–49% for denoising with typical PSNR decrease <0.4 dB (Table 4), and by ~4%–20% for super-resolution with maximum PSNR drop <0.2 dB (Table 5). This demonstrates a consistent and controllable efficiency–performance trade-off.

---

## On mechanistic understanding of high-dimensional residual refinement

- These analyses support high-dimensional residual refinement as the core benefit of the Hourglass design. Our hypothesis is that increasing the latent dimension (**$d_z$**) enlarges the ambient signal space in which the residual path operates, enabling more expressive and separable feature learning.
- **Convergence dynamics**: On ImageNet-224×224, under similar trainable parameters, Hourglass consistently reached lower loss and higher PSNR earlier in training compared to the baseline MLP, indicating faster convergence (Table 6).
- **Effective rank analysis**: Larger **$d_z$** yields higher effective rank (Roy & Vetterli, 2007) and smaller RMS L2 distance between first- and last-layer activations (Table 7), suggesting more compact yet more separable feature spaces that allow residual learning to refine representations along shorter “paths.”
- **Hourglass vs. conventional MLP**: At matched parameter counts, Hourglass achieves both higher effective rank and better PSNR than a conventional MLP, even when the latter has a large ambient dimension (Table 8), consistent with the theoretical advantage of operating residual updates in a wider latent space.

---

> ### Author Response · Authors · 2025-12-02
> **Summary comment to the Area Chair (cont.)**
>
> ## On the absence of Vision Transformer / MLP-Mixer experiments
>
> - Our work focuses on generative tasks (e.g., denoising, super-resolution, language modeling), rather than classification. While Vision Transformers (ViT) and MLP-Mixers are important architectures, they are primarily designed and benchmarked for classification tasks. To evaluate Hourglass in a generative Transformer setting, we chose OLMo2 — a 2025 large language model — as a modern testbed aligned with our scope.
> - This choice was motivated by: (1) architectural relevance (Transformer-based generative model); and (2) scale and representativeness (1B-parameter scale, substantially larger than the largest ViT in the original 2020 paper at ~632M). We believe demonstrating feasibility and potential in a modern, large-scale generative Transformer is more representative of the challenges and trade-offs in current high-dimensional architectures.
> - We appreciate the reviewer’s suggestion to explore Vision Transformers and diffusion models, and we plan to extend our evaluation to these architectures in future work to further validate the generality of the Hourglass design.
>
> ---
>
> ## On novelty and positioning
>
> - To the best of our knowledge, this is the first work to introduce the Wide–Narrow–Wide residual design directly into pure MLP feed-forward layers—with residual updates operating in a higher-dimensional space than the main hidden layer—and the first to apply such Hourglass FFNs in large language models. This differs from U-Net skips, MoE “temporary widenings,” or LoRA-style paths, as our residual path consistently operates at the wider dimensionality and is explicitly optimized for the performance–parameter trade-off.
>
> ---
>
> ## On interoperability with attention in Transformers
>
> - In OLMo2 1B, we widened the residual space (**$d_z$**) while narrowing the bottleneck (**$d_h$**), offsetting the cost and preventing compute from exploding. The Hourglass design interoperates with attention without increasing computational cost, while maintaining competitive early-stage performance (Table 1).
>
> ---
>
> ## On consistency and reporting clarifications
>
> - We clarify that the ImageNet-32 search used **$d_z ∈ [3072, 4576]$**, consistent with the input dimension. We used 5-sigma in some figures purely for visualization, and will adopt the standard 1-sigma reporting in the revised version.
>
> ---
>
> ## Summary and integration into the final version
>
> - We extended evaluation to high-resolution vision tasks (ImageNet-224×224) and to a large-scale generative Transformer (OLMo2 1B), where Hourglass remains on the Pareto frontier (Table 2, Table 3) and achieves similar early training performance with 22% fewer parameters (Table 1).
> - Fixing **$W_{in}$** improves parameter efficiency by reducing trainable parameters (and optimizer state) without changing total parameters or inference complexity, with small performance trade-offs (Table 4, Table 5).
> - Mechanistically, Hourglass shows faster convergence (Table 6), higher effective-rank representations and more compact refinement paths (Table 7), and superior PSNR at matched parameter counts (Table 8).
> - All added experiments and analyses (Tables 1–8) will be integrated into the final version to make the work more complete. We appreciate the reviewers’ calls for broader coverage (e.g., ViT/MLP-Mixer) and, while our focus is generative tasks, we plan to extend to these architectures in future work to further validate generality.

---

> ### Author Response · Authors · 2025-12-02
> **Experiment tables**
>
> ## Table 1. Comparison between original OLMo2 1B and OLMo2 1B-Hourglass on 20B token training
> Both models are trained from scratch under identical settings.
>
>
>
> | Model                  | Params (B) | $d_z$  | $d_h$  | $L$  | Train Loss | Perplexity | Arc Easy |
> |------------------------|------------|------|------|----|------------|------------|----------|
> | OLMo2 1B (baseline)    | 1.484      | 2048 | 8192 | 16 | 2.849      | 17.27      | 62.63    |
> | OLMo2 1B-Hourglass     | 1.153      | 2560 | 1792 | 16 | 2.941      | 18.94      | 61.75    |
>
> ---
>
> ## Table 2. ImageNet 224×224 Denoising — Conventional vs. Hourglass
>
>
>
> | Architecture  | Total Params (M) | $d_z$  | $d_h$  | $L$  | PSNR   |
> |---------------|------------------|------|------|----|--------|
> | Conventional  | 37.767 | 3072 | 3075 | 1 | 23.210 |
> | Conventional  | 46.989 | 3072 | 4576 | 1 | 23.210 |
> | Conventional  | 62.448 | 3072 | 3546 | 2 | 24.704 |
> | Conventional  | 68.174 | 3072 | 4012 | 2 | 24.724 |
> | Conventional  | 75.104 | 3072 | 4576 | 2 | 24.743 |
> | Conventional  | 75.553 | 3072 | 3075 | 3 | 24.939 |
> | Conventional  | 84.234 | 3072 | 3546 | 3 | 24.958 |
> | Hourglass     | 19.286 | 3075 | 16   | 4  | 23.592 |
> | Hourglass     | 19.877 | 3075 | 16   | 10 | 23.767 |
> | Hourglass     | 20.861 | 3075 | 16   | 20 | 23.961 |
> | Hourglass     | 21.722 | 3075 | 115  | 4  | 24.065 |
> | Hourglass     | 22.429 | 3075 | 115  | 5  | 24.172 |
> | Hourglass     | 23.136 | 3075 | 115  | 6  | 24.272 |
> | Hourglass     | 23.844 | 3075 | 115  | 7  | 24.347 |
> | Hourglass     | 23.874 | 3075 | 270  | 3  | 24.359 |
> | Hourglass     | 25.535 | 3075 | 270  | 4  | 24.446 |
> | Hourglass     | 28.856 | 3075 | 270  | 6  | 24.627 |
> | Hourglass     | 33.007 | 3075 | 765  | 3  | 24.643 |
> | Hourglass     | 37.712 | 3075 | 765  | 4  | 24.780 |
> | Hourglass     | 42.417 | 3075 | 765  | 5  | 24.859 |
> | Hourglass     | 47.084 | 3075 | 1146 | 4  | 24.878 |
> | Hourglass     | 47.121 | 3075 | 765  | 6  | 24.901 |
> | Hourglass     | 54.339 | 3546 | 765  | 6  | 24.925 |
> | Hourglass     | 61.480 | 4012 | 765  | 6  | 24.942 |
> | Hourglass     | 66.041 | 3546 | 1560 | 4  | 24.959 |
> | Hourglass     | 87.237 | 4012 | 1560 | 5  | 25.025 |
>
>
>
> ---
>
>
>
> ## Table 3. ImageNet 224×224 Super-resolution — Conventional vs. Hourglass
>
>
>
> | Architecture  | Total Params (M) | $d_z$  | $d_h$  | $L$  | PSNR   |
> |---------------|------------------|------|------|----|--------|
> | Conventional  | 30.689 | 3072 | 3075 | 1 | 26.785 |
> | Conventional  | 33.583 | 3072 | 3546 | 1 | 26.787 |
> | Conventional  | 49.582 | 3072 | 3075 | 2 | 27.836 |
> | Conventional  | 68.475 | 3072 | 3075 | 3 | 28.030 |
> | Conventional  | 77.156 | 3072 | 3546 | 3 | 28.030 |
> | Conventional  | 87.368 | 3072 | 3075 | 4 | 28.084 |
> | Hourglass     | 11.906 | 3075 | 16 | 1 | 26.548 |
> | Hourglass     | 12.005 | 3075 | 16 | 2 | 26.968 |
> | Hourglass     | 12.103 | 3075 | 16 | 3 | 27.155 |
> | Hourglass     | 12.202 | 3075 | 16 | 4 | 27.242 |
> | Hourglass     | 12.300 | 3075 | 16 | 5 | 27.301 |
> | Hourglass     | 12.694 | 3075 | 48 | 3 | 27.395 |
> | Hourglass     | 12.989 | 3075 | 48 | 4 | 27.448 |
> | Hourglass     | 13.284 | 3075 | 48 | 5 | 27.475 |
> | Hourglass     | 14.453 | 3075 | 86 | 5 | 27.504 |
> | Hourglass     | 14.637 | 3075 | 115 | 4 | 27.516 |
> | Hourglass     | 14.981 | 3075 | 86 | 6 | 27.521 |
> | Hourglass     | 15.510 | 3075 | 86 | 7 | 27.556 |
> | Hourglass     | 16.039 | 3075 | 86 | 8 | 27.593 |
> | Hourglass     | 16.568 | 3075 | 86 | 9 | 27.632 |
> | Hourglass     | 17.097 | 3075 | 86 | 10 | 27.668 |
> | Hourglass     | 17.466 | 3075 | 115 | 8 | 27.696 |
> | Hourglass     | 18.450 | 3075 | 270 | 4 | 27.723 |
> | Hourglass     | 18.881 | 3075 | 115 | 10 | 27.763 |
> | Hourglass     | 21.771 | 3075 | 270 | 6 | 27.860 |
> | Hourglass     | 25.092 | 3075 | 270 | 8 | 27.920 |
> | Hourglass     | 28.935 | 3546 | 270 | 8 | 27.931 |
> | Hourglass     | 30.627 | 3075 | 765 | 4 | 27.965 |
> | Hourglass     | 35.318 | 3546 | 765 | 4 | 27.980 |
> | Hourglass     | 40.000 | 3075 | 1146 | 4 | 28.018 |
> | Hourglass     | 46.126 | 3546 | 1146 | 4 | 28.031 |
> | Hourglass     | 46.169 | 3546 | 765 | 6 | 28.049 |
> | Hourglass     | 57.871 | 3546 | 1560 | 4 | 28.069 |
> | Hourglass     | 61.352 | 3075 | 2014 | 4 | 28.077 |
> | Hourglass     | 69.372 | 3075 | 1560 | 6 | 28.086 |
> | Hourglass     | 70.750 | 3546 | 2014 | 4 | 28.092 |
> | Hourglass     | 80.047 | 4012 | 2014 | 4 | 28.104 |

---

> ### Author Response · Authors · 2025-12-02
> **Experiment tables**
>
> ## Table 4. ImageNet 224×224 Denoising — Hourglass vs. Hourglass (fix $W_{in}$)
>
>
>
> | Architecture                | Total Params (M) | Trainable Params (M) | $d_z$  | $d_h$  | $L$  | PSNR   |
> |-----------------------------|------------------|----------------------|------|------|----|--------|
> | Hourglass                   | 19.286 | 19.286 | 3075 | 16   | 4  | 23.592 |
> | Hourglass                   | 19.877 | 19.877 | 3075 | 16   | 10 | 23.767 |
> | Hourglass                   | 20.861 | 20.861 | 3075 | 16   | 20 | 23.961 |
> | Hourglass                   | 21.722 | 21.722 | 3075 | 115  | 4  | 24.065 |
> | Hourglass                   | 22.429 | 22.429 | 3075 | 115  | 5  | 24.172 |
> | Hourglass                   | 23.136 | 23.136 | 3075 | 115  | 6  | 24.272 |
> | Hourglass                   | 23.844 | 23.844 | 3075 | 115  | 7  | 24.347 |
> | Hourglass                   | 23.874 | 23.874 | 3075 | 270  | 3  | 24.359 |
> | Hourglass                   | 25.535 | 25.535 | 3075 | 270  | 4  | 24.446 |
> | Hourglass                   | 28.856 | 28.856 | 3075 | 270  | 6  | 24.627 |
> | Hourglass                   | 33.007 | 33.007 | 3075 | 765  | 3  | 24.643 |
> | Hourglass                   | 37.712 | 37.712 | 3075 | 765  | 4  | 24.780 |
> | Hourglass                   | 42.417 | 42.417 | 3075 | 765  | 5  | 24.859 |
> | Hourglass                   | 47.084 | 47.084 | 3075 | 1146 | 4  | 24.878 |
> | Hourglass                   | 47.121 | 47.121 | 3075 | 765  | 6  | 24.901 |
> | Hourglass                   | 54.339 | 54.339 | 3546 | 765  | 6  | 24.925 |
> | Hourglass                   | 61.480 | 61.480 | 4012 | 765  | 6  | 24.942 |
> | Hourglass                   | 66.041 | 66.041 | 3546 | 1560 | 4  | 24.959 |
> | Hourglass                   | 87.237 | 87.237 | 4012 | 1560 | 5  | 25.025 |
> | Hourglass (fix $W_{in}$)        | 19.286 | 9.840  | 3075 | 16   | 4  | 23.215 |
> | Hourglass (fix $W_{in}$)        | 19.877 | 10.430 | 3075 | 16   | 10 | 23.392 |
> | Hourglass (fix $W_{in}$)        | 20.861 | 11.414 | 3075 | 16   | 20 | 23.487 |
> | Hourglass (fix $W_{in}$)        | 21.722 | 12.275 | 3075 | 115  | 4  | 23.783 |
> | Hourglass (fix $W_{in}$)        | 22.429 | 12.983 | 3075 | 115  | 5  | 23.869 |
> | Hourglass (fix $W_{in}$)        | 23.136 | 13.690 | 3075 | 115  | 6  | 23.956 |
> | Hourglass (fix $W_{in}$)        | 23.844 | 14.397 | 3075 | 115  | 7  | 24.015 |
> | Hourglass (fix $W_{in}$)        | 23.874 | 14.428 | 3075 | 270  | 3  | 24.074 |
> | Hourglass (fix $W_{in}$)        | 25.535 | 16.088 | 3075 | 270  | 4  | 24.134 |
> | Hourglass (fix $W_{in}$)        | 28.856 | 19.409 | 3075 | 270  | 6  | 24.293 |
> | Hourglass (fix $W_{in}$)        | 33.007 | 23.561 | 3075 | 765  | 3  | 24.332 |
> | Hourglass (fix $W_{in}$)        | 37.712 | 28.265 | 3075 | 765  | 4  | 24.452 |
> | Hourglass (fix $W_{in}$)        | 42.417 | 32.970 | 3075 | 765  | 5  | 24.533 |
> | Hourglass (fix $W_{in}$)        | 47.084 | 37.638 | 3075 | 1146 | 4  | 24.541 |
> | Hourglass (fix $W_{in}$)        | 47.121 | 37.675 | 3075 | 765  | 6  | 24.560 |
> | Hourglass (fix $W_{in}$)        | 54.339 | 43.446 | 3546 | 765  | 6  | 24.598 |
> | Hourglass (fix $W_{in}$)        | 61.480 | 49.155 | 4012 | 765  | 6  | 24.691 |
> | Hourglass (fix $W_{in}$)        | 66.041 | 55.147 | 3546 | 1560 | 4  | 24.701 |
> | Hourglass (fix $W_{in}$)        | 87.237 | 74.912 | 4012 | 1560 | 5  | 24.799 |
>
>
>
> ---

---

> ### Author Response · Authors · 2025-12-02
> **Experiment tables**
>
> ## Table 5. ImageNet 224×224 Super-resolution — Hourglass vs. Hourglass (fix $W_{in}$)
>
> | Architecture                | Total Params (M) | Trainable Params (M) | $d_z$  | $d_h$  | $L$  | PSNR   |
> |-----------------------------|------------------|----------------------|------|------|----|--------|
> | Hourglass                   | 11.906 | 11.906 | 3075 | 16 | 1 | 26.548 |
> | Hourglass                   | 12.005 | 12.005 | 3075 | 16 | 2 | 26.968 |
> | Hourglass                   | 12.103 | 12.103 | 3075 | 16 | 3 | 27.155 |
> | Hourglass                   | 12.202 | 12.202 | 3075 | 16 | 4 | 27.242 |
> | Hourglass                   | 12.300 | 12.300 | 3075 | 16 | 5 | 27.301 |
> | Hourglass                   | 12.694 | 12.694 | 3075 | 48 | 3 | 27.395 |
> | Hourglass                   | 12.989 | 12.989 | 3075 | 48 | 4 | 27.448 |
> | Hourglass                   | 13.284 | 13.284 | 3075 | 48 | 5 | 27.475 |
> | Hourglass                   | 14.453 | 14.453 | 3075 | 86 | 5 | 27.504 |
> | Hourglass                   | 14.637 | 14.637 | 3075 | 115 | 4 | 27.516 |
> | Hourglass                   | 14.982 | 14.981 | 3075 | 86 | 6 | 27.521 |
> | Hourglass                   | 15.510 | 15.510 | 3075 | 86 | 7 | 27.556 |
> | Hourglass                   | 16.039 | 16.039 | 3075 | 86 | 8 | 27.593 |
> | Hourglass                   | 16.568 | 16.568 | 3075 | 86 | 9 | 27.632 |
> | Hourglass                   | 17.097 | 17.097 | 3075 | 86 | 10 | 27.668 |
> | Hourglass                   | 17.466 | 17.466 | 3075 | 115 | 8 | 27.696 |
> | Hourglass                   | 18.450 | 18.450 | 3075 | 270 | 4 | 27.723 |
> | Hourglass                   | 18.881 | 18.881 | 3075 | 115 | 10 | 27.763 |
> | Hourglass                   | 21.771 | 21.771 | 3075 | 270 | 6 | 27.860 |
> | Hourglass                   | 25.092 | 25.092 | 3075 | 270 | 8 | 27.920 |
> | Hourglass                   | 28.935 | 28.935 | 3546 | 270 | 8 | 27.931 |
> | Hourglass                   | 30.627 | 30.627 | 3075 | 765 | 4 | 27.965 |
> | Hourglass                   | 35.318 | 35.318 | 3546 | 765 | 4 | 27.980 |
> | Hourglass                   | 40.000 | 40.000 | 3075 | 1146 | 4 | 28.018 |
> | Hourglass                   | 46.126 | 46.126 | 3546 | 1146 | 4 | 28.031 |
> | Hourglass                   | 46.169 | 46.169 | 3546 | 765 | 6 | 28.049 |
> | Hourglass                   | 57.871 | 57.871 | 3546 | 1560 | 4 | 28.069 |
> | Hourglass                   | 61.352 | 61.352 | 3075 | 2014 | 4 | 28.077 |
> | Hourglass                   | 69.372 | 69.372 | 3075 | 1560 | 6 | 28.086 |
> | Hourglass                   | 70.750 | 70.750 | 3546 | 2014 | 4 | 28.092 |
> | Hourglass                   | 80.047 | 80.047 | 4012 | 2014 | 4 | 28.104 |
> | Hourglass (fix W_in)        | 11.906 | 9.545  | 3075 | 16 | 1 | 26.445 |
> | Hourglass (fix W_in)        | 12.005 | 9.643  | 3075 | 16 | 2 | 26.826 |
> | Hourglass (fix W_in)        | 12.103 | 9.742  | 3075 | 16 | 3 | 26.992 |
> | Hourglass (fix W_in)        | 12.202 | 9.840  | 3075 | 16 | 4 | 27.041 |
> | Hourglass (fix W_in)        | 12.300 | 9.938  | 3075 | 16 | 5 | 27.094 |
> | Hourglass (fix W_in)        | 12.694 | 10.332 | 3075 | 48 | 3 | 27.120 |
> | Hourglass (fix W_in)        | 12.989 | 10.627 | 3075 | 48 | 4 | 27.165 |
> | Hourglass (fix W_in)        | 13.284 | 10.922 | 3075 | 48 | 5 | 27.191 |
> | Hourglass (fix W_in)        | 14.453 | 12.091 | 3075 | 86 | 5 | 27.321 |
> | Hourglass (fix W_in)        | 14.637 | 12.275 | 3075 | 115 | 4 | 27.331 |
> | Hourglass (fix W_in)        | 14.982 | 12.620 | 3075 | 86 | 6 | 27.371 |
> | Hourglass (fix W_in)        | 15.510 | 13.149 | 3075 | 86 | 7 | 27.429 |
> | Hourglass (fix W_in)        | 16.039 | 13.678 | 3075 | 86 | 8 | 27.492 |
> | Hourglass (fix W_in)        | 16.568 | 14.207 | 3075 | 86 | 9 | 27.541 |
> | Hourglass (fix W_in)        | 17.097 | 14.735 | 3075 | 86 | 10 | 27.581 |
> | Hourglass (fix W_in)        | 17.466 | 15.104 | 3075 | 115 | 8 | 27.607 |
> | Hourglass (fix W_in)        | 18.450 | 16.088 | 3075 | 270 | 4 | 27.634 |
> | Hourglass (fix W_in)        | 18.881 | 16.519 | 3075 | 115 | 10 | 27.677 |
> | Hourglass (fix W_in)        | 21.771 | 19.409 | 3075 | 270 | 6 | 27.764 |
> | Hourglass (fix W_in)        | 25.092 | 22.730 | 3075 | 270 | 8 | 27.823 |
> | Hourglass (fix W_in)        | 28.935 | 26.212 | 3546 | 270 | 8 | 27.833 |
> | Hourglass (fix W_in)        | 30.627 | 28.265 | 3075 | 765 | 4 | 27.852 |
> | Hourglass (fix W_in)        | 35.318 | 32.595 | 3546 | 765 | 4 | 27.870 |
> | Hourglass (fix W_in)        | 40.000 | 37.638 | 3075 | 1146 | 4 | 27.891 |
> | Hourglass (fix W_in)        | 46.126 | 43.403 | 3546 | 1146 | 4 | 27.905 |
> | Hourglass (fix W_in)        | 46.169 | 43.446 | 3546 | 765 | 6 | 27.909 |
> | Hourglass (fix W_in)        | 57.871 | 55.147 | 3546 | 1560 | 4 | 27.918 |
> | Hourglass (fix W_in)        | 61.352 | 58.991 | 3075 | 2014 | 4 | 27.919 |
> | Hourglass (fix W_in)        | 69.372 | 67.010 | 3075 | 1560 | 6 | 27.927 |
> | Hourglass (fix W_in)        | 70.750 | 68.026 | 3546 | 2014 | 4 | 27.928 |
> | Hourglass (fix W_in)        | 80.047 | 76.966 | 4012 | 2014 | 4 | 27.937 |

---

> ### Author Response · Authors · 2025-12-02
> **Experiment tables**
>
> ## Table 6 — Loss and PSNR on evaluation set at different training steps
>
> **Description:**
> Loss and PSNR on the evaluation set at different training steps for two model configurations lying on the Pareto fronts:
> - **Conventional:** $d_z = 3072$, $d_h = 3546$, $L = 2$, **$55.37 M$ params**
> - **Hourglass:** $d_z = 3546$, $d_h = 1146$, $L = 5$, **$54.25 M$ params**
>
> Results show that the Hourglass architecture converges faster compared to the conventional architecture.
>
> | Step   | Conv. Loss | Hourglass Loss | Conv. PSNR | Hourglass PSNR |
> |--------|------------|----------------|------------|----------------|
> | 500    | **0.00685** | 0.00695        | **21.6466** | 21.5820        |
> | 1000   | **0.00609** | 0.00630        | **22.1584** | 22.0090        |
> | 1500   | 0.00579    | **0.00571**     | 22.3799    | **22.4381**    |
> | 2000   | 0.00569    | **0.00552**     | 22.4505    | **22.5890**    |
> | 2500   | 0.00566    | **0.00538**     | 22.4759    | **22.7012**    |
> | 5000   | 0.00514    | **0.00493**     | 22.8949    | **23.0802**    |
> | 7500   | 0.00486    | **0.00467**     | 23.1399    | **23.3118**    |
> | 10000  | 0.00462    | **0.00448**     | 23.3552    | **23.4951**    |
> | 12500  | 0.00450    | **0.00434**     | 23.4768    | **23.6278**    |
> | 15000  | 0.00432    | **0.00420**     | 23.6473    | **23.7769**    |
> | 17500  | 0.00421    | **0.00410**     | 23.7599    | **23.8737**    |
> | 20000  | 0.00411    | **0.00401**     | 23.8686    | **23.9713**    |
> ---
> ## Table 7 — Effective rank analysis of Hourglass model
>
> **Description:**
> Effective rank of Hourglass model with $d_z \in$ {3075, 3546, 4012, 4576}, $d_h = 1146$, $L = 5$, using fixed learning rate $3e-4$.
>
> | $d_z$   | PSNR     | Effective Rank | RMS($h_L − h_1$) ± std |
> |-------|----------|----------------|----------------------|
> | 3075  | 23.9702  | 1500.73        | 0.6376 ± 0.0939       |
> | 3546  | 23.9823  | 1699.80        | 0.6205 ± 0.0860       |
> | 4012  | 23.9857  | 1871.54        | 0.5988 ± 0.0883       |
> ---
> ## Table 8 — Hourglass vs. Conventional MLP (Effective Rank and PSNR)
>
> | Architecture | $d_z$   | $d_h$   | $L$  | Total Params (M) | PSNR      | Effective Rank |
> |--------------|-------|-------|----|------------------|-----------|----------------|
> | Conventional | 3072  | 3546  | 2  | 55.37            | 23.8783   | 1180.63        |
> | Hourglass    | 3075  | 1146  | 6  | 55.00            | **23.9701** | **1501.27**    |

---

### Meta-Review · Area_Chair_Ek12 · 2026-01-07

**Summary:**

This work proposes using an hourglass network with wide features in skip connections or MLP layers as an efficient alternative to the narrow–wide–narrow convention.
All reviewers evaluated it on the rejection side, mostly because of little empirical support.

All reviewers evaluated it on the rejection side, mostly because of little empirical support.
This work scopes the proposition of a practical architecture without theoretical backing, and it requires comprehensive experiments, but the proposed architecture is explored only on limited models and datasets (i.e., MNIST and ImageNet-32). Although the authors added several experiments in the rebuttal, that is, ImageNet-224 and a transformer architecture (OLMo2), there still seems to be significant room to explore.
For example, the single transformer experiment was conducted only with the same hyperparameter settings as the conventional baseline, which makes the claim of Pareto optimality less convincing. Adding more variants of modern architectures that reviewers recommended, such as MLP-Mixer and Vision Transformers, would further increase the evidence to support the current concept.
Thus, I also evaluate the current work as a rejection.

**Reviewer Concerns:**

In particular, the following concerns remain outstanding:

**Reviewer ryCo**
- The experiments of the paper are too narrow and simple, ... This is in contrast to the high-dimensional, large-scale settings for modern LLMs and diffusion models, where architectural design trade-offs make a large difference on both performance and efficiency.

**Reviewer rEMH**
- Pareto-Optimal with no theoretical backing. Only based on empirical results from a small subset of conventional models
- The current experiments are limited it would be more convincing to compared on more computer vision task like ... Bottleneck-Resnet and MLPMixer

**Reviewer cH7F**
- Evaluation scope & measures do not support ambition. The experiment tasks are small-data, low-res, PSNR only. Improvements at high-budget end are small

**Reviewer Scores:**

The reviewers are fairly confident on their claims and I expect no score changes.

---

### Decision · Program_Chairs · 2026-01-26

Reject